# Only Strict Saddles in the Energy Landscape of Predictive Coding Networks?

**Francesco Innocenti**
School of Engineering and Informatics
University of Sussex
F.Innocenti@sussex.ac.uk

**El Mehdi Achour**
RWTH Aachen University
Aachen, Germany
achour@mathc.rwth-aachen.de

**Ryan Singh**
School of Engineering and Informatics
University of Sussex
rs773@sussex.ac.uk

**Christopher L. Buckley**
School of Engineering and Informatics
University of Sussex
VERSES
c.l.buckley@sussex.ac.uk

## Abstract

Predictive coding (PC) is an energy-based learning algorithm that performs iterative inference over network activities before updating weights. Recent work suggests that PC can converge in fewer learning steps than backpropagation thanks to its inference procedure. However, these advantages are not always observed, and the impact of PC inference on learning is not theoretically well understood. To address this gap, we study the geometry of the PC weight landscape at the inference equilibrium of the network activities. For deep linear networks, we first show that the equilibrated PC energy is equal to a rescaled mean squared error loss with a weight-dependent rescaling. We then prove that many highly degenerate (non-strict) saddles of the loss including the origin become much easier to escape (strict) in the equilibrated energy. Experiments on both linear and non-linear networks strongly validate our theory and further suggest that all the saddles of the equilibrated energy are strict. Overall, this work shows that PC inference makes the loss landscape of feedforward networks more benign and robust to vanishing gradients, while also highlighting the fundamental challenge of scaling PC to very deep models.

## 1 Introduction

Originating as a general principle of brain function, predictive coding (PC) has in recent years been developed into a local learning algorithm that could provide a biologically plausible alternative to backpropagation (BP) [32, 31, 43]. Deep neural networks (DNNs) trained with PC have shown comparable performance to BP on standard small-to-medium machine learning tasks, including classification, generation and memory association [31, 43, 41]. PC networks (PCNs) are also highly versatile, allowing for arbitrary computational graphs [45, 10], hybrid and causal inference [44, 59], and temporal prediction [35].

In contrast to BP, and similar to other energy-based algorithms [e.g. 49, 38], PC performs iterative (approximately Bayesian) inference over network activities before weight updates. This has been recently described as a fundamentally different principle of credit assignment for learning in the brain called "prospective configuration" [54], where weights follow activities (rather than the other way around). While the inference process key to PC incurs an additional computational cost, it has been suggested to provide many benefits for learning including faster convergence [54, 3, 18]. However,

38th Conference on Neural Information Processing Systems (NeurIPS 2024).

these speed-ups are not consistently observed across datasets, models and optimisers [3], and the impact of PC inference on learning more generally is not theoretically well understood (see §A.2.1).

To address this gap, we study the geometry of the effective landscape on which PC learns: *the weight landscape at the inference equilibrium of the network activities* (defined in §2.2). Our theory considers deep linear networks (DLNs), the standard model for theoretical studies of the loss landscape (see §A.2). Despite being able to learn only linear representations, DLNs have non-convex loss landscapes with non-linear learning dynamics that have proved to be a useful model for understanding non-linear networks [e.g. 48]. In contrast to previous theories of PC [3, 2, 18], we do not make any additional assumptions or approximations (see §A.2), and we empirically verify that our linear theory holds for non-linear networks.

For DLNs, we first show that, at the inference equilibrium, the PC energy is equal to a rescaled mean squared error (MSE) loss with a non-trivial, weight-dependent rescaling (Theorem 1). We then compare saddle points of the loss, which have been recently characterised [23, 1], to those of the equilibrated energy. Such saddles, which are ubiquitous in the loss landscape of neural networks [11, 1], can be of two main types: "strict", where the Hessian is indefinite (Def. 1); and "non-strict", where an escape direction is found in higher-order derivatives [15, 23, 1]. Non-strict saddles are particularly problematic for first-order methods like (stochastic) gradient descent (SGD) since they are by definition at least second-order critical points. While SGD can be exponentially slowed in the vicinity of strict saddles [12], it can effectively get stuck in non-strict ones [47, 7] (see §A.2 for a review). This is the phenomenon of vanishing gradients viewed from a landscape perspective [39, 6].

By contrast, here we prove that many non-strict saddles of the MSE loss, specifically saddles of rank zero, become strict in the equilibrated energy of any DLN (Theorems 2 & 3). These saddles include the origin, whose degeneracy (i.e. flatness) in the loss grows with the number of hidden layers. Our theoretical results are strongly validated by experiments on both linear and non-linear networks, and additional experiments suggest that other (higher-rank) non-strict saddles of the loss are strict in the equilibrated energy. Based on these results, we conjecture that all the saddles of the equilibrated energy are strict. Overall, this work suggests that PC inference makes the loss landscape more benign and robust to vanishing gradients, while also highlighting the fundamental challenge of speeding up PC inference on deeper networks.

The rest of the paper is structured as follows. After introducing the setup (§2), we present our theoretical results for DLNs (§3), including some illustrative examples and thorough empirical verifications of each result. We then report experiments on non-linear networks supporting our theory and more general conjecture (§4). We conclude by discussing the implications and limitations of our work, as well as potential future directions (§5). Appendix A includes a review of related work, derivations, experiment details and supplementary results. Code to reproduce all the experiments is available at https://github.com/francesco-innocenti/pc-saddles.

## 1.1 Summary of contributions

- We derive an exact solution for the PC energy of DLNs at the inference equilibrium (Theorem 1), which turns out to be a rescaled MSE loss with a weight-dependent rescaling. This corrects a previous mistake in the literature that the MSE loss is equal to the output energy [34] (which holds only at the feedforward pass) and enables further studies of the PC energy landscape. We find an excellent match between our theory and experiment (Figure 1).

- Based on this result, we prove that, in contrast to the MSE, the origin of the equilibrated energy of DLNs is a strict saddle independent of network depth. We provide an explicit characterisation of the Hessian at the origin of the equilibrated energy (Theorem 2), which is perfectly validated by experiments on linear networks (Figures 3, 4 & 8).

- We further prove that other non-strict saddles of the MSE than the origin, specifically saddles of rank zero, become strict in the equilibrated energy of DLNs (Theorem 3). We provide an empirical verification of one of these saddles as an example (Figures 9 & 10).

- We empirically show that our linear theory holds for non-linear networks, including convolutional architectures, trained on standard image classification tasks. In particular, when initialised close to non-strict saddles of the MSE covered by Theorem 3, we find that SGD on the equilibrated energy escapes much faster than on the loss given the same learning rate (Figures 5 & 12). In contrast to BP, PC exhibits no vanishing gradients (Figure 11).

- We perform additional experiments, again on both linear and non-linear networks, showing that PC quickly escapes other (higher-rank) non-strict saddles of the MSE that we do not address theoretically (Figure 6), supporting our conjecture that all the saddles of the equilibrated energy are strict.

## 2 Preliminaries

**Notation.** We use the following shorthand $\mathbf{W}_{k:\ell} = \mathbf{W}_k \ldots \mathbf{W}_\ell$ for $\ell, k \in 1, \ldots, L$, denoting the total product of weight matrices as $\mathbf{W}_{L:1} = \mathbf{W}_L \ldots \mathbf{W}_1$. $\mathbf{I}_n$ is the $n \times n$ identity matrix, while $\mathbf{0}_n$ denotes either the $n$-zero vector or the $n \times n$ null matrix, and $n$ will be omitted when clear from context. $||\cdot||$ denotes the $\ell_2$ norm, and $\otimes$ is the Kronecker product between two matrices. We will consider the gradient and Hessian of an objective $f$ only with respect to the network weights $\boldsymbol{\theta}$ and sometimes abbreviate them as $\mathbf{g}_f := \nabla_{\boldsymbol{\theta}} f$ and $\mathbf{H}_f := \nabla_{\boldsymbol{\theta}}^2 f$, respectively, omitting the independent variable for simplicity. The largest and smallest eigenvalues of the Hessian are $\lambda_{\max}(\mathbf{H}_f)$ and $\lambda_{\min}(\mathbf{H}_f)$, with $\hat{\mathbf{v}}_{\max}$ and $\hat{\mathbf{v}}_{\min}$ as their associated eigenvectors. See §A.1 for more general notation.

**Definition 1.** *Strict saddle.* Following [15] and later work, any critical point $\boldsymbol{\theta}^*$ of $f(\boldsymbol{\theta})$ where $\mathbf{g}_f(\boldsymbol{\theta}^*) = \mathbf{0}$ is defined as a strict saddle when the Hessian at that point has at least one negative eigenvalue, $\lambda_{\min}(\mathbf{H}_f(\boldsymbol{\theta}^*)) < 0$. Any other critical point with a positive semi-definite Hessian and at least one negative eigenvalue in a higher-order derivative is said to be a non-strict saddle.

### 2.1 Deep Linear Networks (DLNs)

We consider DLNs with one or more hidden layers $H = L - 1 \geq 1$ defining the linear mapping $\mathbf{W}_{L:1} : \mathbb{R}^{d_x} \to \mathbb{R}^{d_y}$ where $\mathbf{W}_\ell \in \mathbb{R}^{n_\ell \times n_{\ell-1}}$, with layer widths $\{n_\ell\}_{\ell=0}^{L}$ and input and output dimensions $n_0 = d_x, n_L = d_y$. We ignore biases for simplicity. The standard MSE loss for DLNs can then be written as

$$\mathcal{L} = \frac{1}{2N} \sum_{i=1}^{N} ||\mathbf{y}_i - \mathbf{W}_{L:1}\mathbf{x}_i||^2, \tag{1}$$

for a dataset of $N$ examples $\{(\mathbf{x}_i, \mathbf{y}_i)\}_{i=1}^{N}$ where $\mathbf{x} \in \mathbb{R}^{d_x}, \mathbf{y} \in \mathbb{R}^{d_y}$. The total number of weights is given by $p = \sum_{\ell=1}^{L} n_\ell n_{\ell-1}$, and we will denote the set of all network parameters as $\boldsymbol{\theta} := \text{vec}(\mathbf{W}_1, \ldots, \mathbf{W}_L) \in \mathbb{R}^p$. For brevity, we will often refer to the MSE loss as simply the loss.

### 2.2 Predictive coding (PC)

DNNs trained with PC typically assume a hierarchical Gaussian model with identity covariances, so we will adopt this formulation for linear fully connected layers $\mathbf{z}_\ell \sim \mathcal{N}(\mathbf{W}_\ell \mathbf{z}_{\ell-1}, \mathbf{I}_\ell)$ where the mean activity of each layer $\mathbf{z}_\ell$ is a linear function of the previous layer. Under some other common assumptions about the generative model, we can derive an energy function, often referred to as the variational free energy, which is a sum of squared prediction errors across layers [9]:

$$\mathcal{F} = \frac{1}{2N} \sum_{i=1}^{N} \sum_{\ell=1}^{L} ||\mathbf{z}_{\ell,i} - \mathbf{W}_\ell \mathbf{z}_{\ell-1,i}||^2. \tag{2}$$

Note that this objective defines an energy for every neuron, highlighting the locality of the algorithm. To train a PCN, the last layer is clamped to some data, $\mathbf{z}_{L,i} := \mathbf{y}_i$, which could be a label for classification or an image for generation. In a supervised task, the first layer is also fixed to some input, $\mathbf{z}_{0,i} := \mathbf{x}_i$. The energy (Eq. 2) is then minimised in two phases, first w.r.t. the activities (inference) and then w.r.t. the weights (learning):

$$\textit{Inference:} \quad \Delta\mathbf{z}_\ell \propto -\frac{\partial\mathcal{F}}{\partial\mathbf{z}_\ell} \qquad (3) \qquad \textit{Learning:} \quad \Delta\mathbf{W}_\ell \propto -\frac{\partial\mathcal{F}}{\partial\mathbf{W}_\ell} \qquad (4)$$

where we omit the data index $i$ for simplicity. In practice, the inference dynamics (Eq. 3) are often run to convergence until $\Delta\mathbf{z}_\ell \approx 0$, before performing a weight (e.g. GD) update (Eq. 4). Importantly, the effective weight landscape on which PC learns is therefore the energy at the inference equilibrium $\mathcal{F}|_{\Delta\mathbf{z}\approx\mathbf{0}}(\boldsymbol{\theta})$, which we will refer to as the equilibrated energy or sometimes simply the energy.

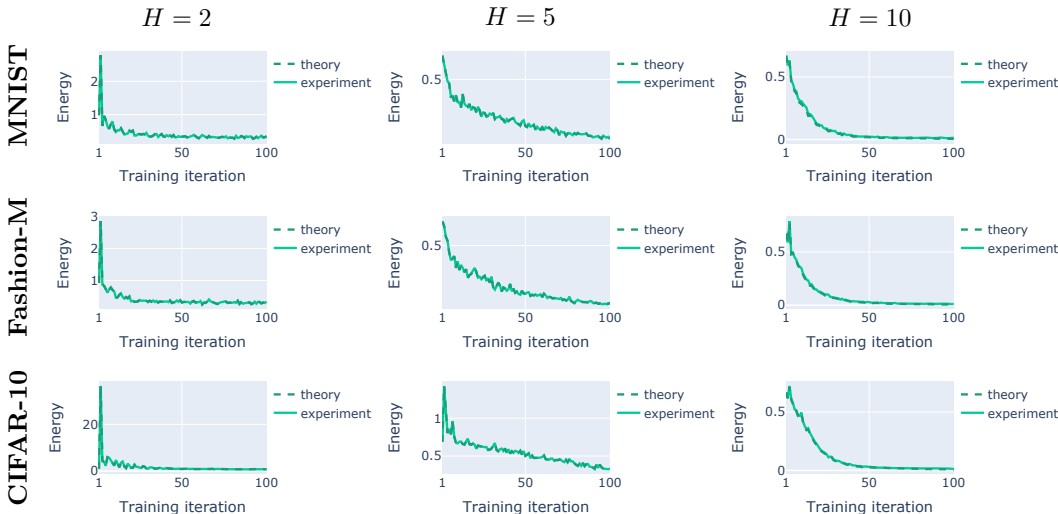

Figure 1: **Empirical verification of the theoretical equilibrated energy of deep linear networks (Theorem 1).** For different datasets, we plot the energy (Eq. 2) at the numerical inference equilibrium $\mathcal{F}|_{\partial\mathcal{F}/\partial\mathbf{z}\approx 0}$ for DLNs with different number of hidden layers $H \in \{2, 5, 10\}$ (see §A.4 for more details), observing an excellent match with the theoretical prediction (Eq. 5).

# 3 Theoretical results

## 3.1 Equilibrated energy as rescaled MSE

As explained in §2.2, the weights of a PCN are typically updated once the activities have converged to an equilibrium. The equilibrated energy $\mathcal{F}|_{\partial\mathcal{F}/\partial\mathbf{z}=0}(\boldsymbol{\theta})$, which we will abbreviate as $\mathcal{F}^*(\boldsymbol{\theta})$, is therefore the effective learning landscape navigated by PC and the object we are interested in studying. It turns out that we can derive a closed-form solution for the equilibrated energy of DLNs, which will be the basis of our subsequent results.

---

**Theorem 1** (Equilibrated energy of DLNs). *For any DLN parameterised by $\boldsymbol{\theta} := (\mathbf{W}_1, \ldots, \mathbf{W}_L)$ with input and output $(\mathbf{x}_i, \mathbf{y}_i)$, the PC energy (Eq. 2) at the exact inference equilibrium $\partial\mathcal{F}/\partial\mathbf{z} = \mathbf{0}$ is the following rescaled MSE loss (see §A.3.2 for derivation)*

$$\mathcal{F}^* = \frac{1}{2N}\sum_{i=1}^{N}(\mathbf{y}_i - \mathbf{W}_{L:1}\mathbf{x}_i)^T\mathbf{S}^{-1}(\mathbf{y}_i - \mathbf{W}_{L:1}\mathbf{x}_i) \tag{5}$$

*where the rescaling is $\mathbf{S} = \mathbf{I}_{d_y} + \sum_{\ell=2}^{L}(\mathbf{W}_{L:\ell})(\mathbf{W}_{L:\ell})^T$.*

---

The proof relies on unfolding the hierarchical Gaussian model assumed by PC to work out the full, implicit generative model of the output, and the rescaling $\mathbf{S}$ comes from the variance modelled by PC at each layer (see §A.3.2 for details). Figure 1 shows an excellent empirical validation of the theory.

Intuitively, the PC inference process (Eq. 3) can then be thought of as reshaping the (MSE) loss landscape to take some layer-wise, weight-dependent variance into account. This immediately raises the question: how does the equilibrated energy landscape $\mathcal{F}^*(\boldsymbol{\theta})$ differ from the loss landscape $\mathcal{L}(\boldsymbol{\theta})$? Is the rescaling—and so the layer variance modelled by PC—useful for learning? Below we provide a partial positive answer to this question by comparing the saddle point geometry of the two objectives.

## 3.2 Analysis of the origin saddle ($\boldsymbol{\theta} = \mathbf{0}$)

Here we prove that, in contrast to the MSE loss, the origin of the equilibrated energy (Eq. 5, where all the weights are zero, $\boldsymbol{\theta} = \mathbf{0}$) is a strict saddle (Def. 1) for DLNs of any depth. To do so, we derive an explicit expression for the Hessian at the origin of the equilibrated energy. For intuitive

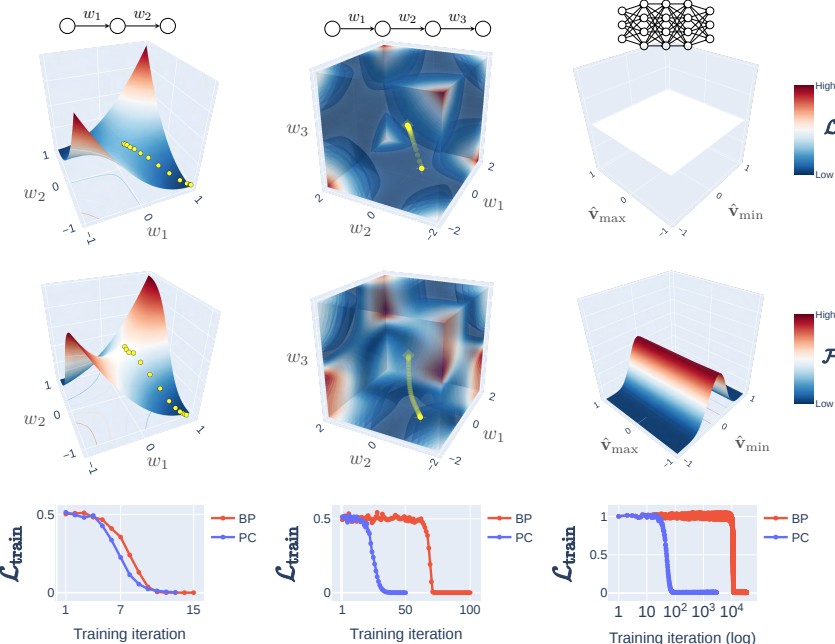

Figure 2: **Toy examples illustrating the (Theorem 2) result that the saddle at the origin of the equilibrated energy is strict independent of network depth.** We plot the MSE loss $\mathcal{L}(\boldsymbol{\theta})$ (*top*) and equilibrated energy landscape $\mathcal{F}^*(\boldsymbol{\theta})$ (*middle*) around the origin for 3 linear networks trained with SGD on a toy problem (see §A.4 for details). We also show the training losses for a representative run with initialisation close to the origin (*bottom*). For the one-dimensional networks, we visualise the landscape around the origin as well as the SGD updates. For the wide network, we project the landscape onto the maximum and minimum eigenvectors of the Hessian, following [7]. Note that in this case the projection of the loss is flat because the Hessian at the origin is zero for $H > 1$ (Eq. 6).

comparison, we first briefly recall the known results that, at the origin, the loss Hessian is indefinite for one-hidden-layer networks and zero for any deeper network (see §A.3.1 for a re-derivation)

$$
\mathbf{H}_{\mathcal{L}}(\boldsymbol{\theta} = \mathbf{0}) = \begin{cases} \begin{bmatrix} \mathbf{0} & -\widetilde{\boldsymbol{\Sigma}}_{\mathbf{xy}} \otimes \mathbf{I}_{n_1} \\ -\mathbf{I}_{n_1} \otimes \widetilde{\boldsymbol{\Sigma}}_{\mathbf{yx}} & \mathbf{0} \end{bmatrix}, & H = 1 \\ \\ \mathbf{0}_p, & H > 1 \end{cases} \tag{6}
$$

where following previous works $\widetilde{\boldsymbol{\Sigma}}_{\mathbf{xy}} := \frac{1}{N} \sum_i^N \mathbf{x}_i \mathbf{y}_i^T$ is the empirical input-output covariance. One-hidden-layer networks $H = 1$ are a special case where the origin saddle of the loss is strict (Def. 1) and was studied in detail by [48] (see left panel of Figure 2 for an example). For deeper networks $H > 1$, the saddle is non-strict as first shown by [23]:

$$
\begin{cases} \lambda_{\min}(\mathbf{H}_{\mathcal{L}}(\boldsymbol{\theta} = \mathbf{0})) < 0, & H = 1 \quad [\text{strict saddle}] \\ \\ \lambda_{\min}(\mathbf{H}_{\mathcal{L}}(\boldsymbol{\theta} = \mathbf{0})) = 0, & H > 1 \quad [\text{non-strict saddle}] \end{cases} \tag{7}
$$

More specifically, the origin saddle of the loss is of order $H^1$, becoming increasingly degenerate (flat) and harder to escape with depth, especially for first-order methods like SGD (see middle and right panels of Figure 2).

By contrast, now we show that the origin saddle of the equilibrated energy is strict for DLNs of any number of hidden layers. Figure 2 shows a few toy examples illustrating the result. In brief, we

---

[1]The $n$th-order of a saddle simply indicates the $(n$th+1) derivative where the first negative (escape) direction is found. So, for example, a first-order (strict) saddle has a zero gradient and an indefinite Hessian, while a second-order (non-strict) saddle has a zero Hessian but a third derivative with a negative direction.

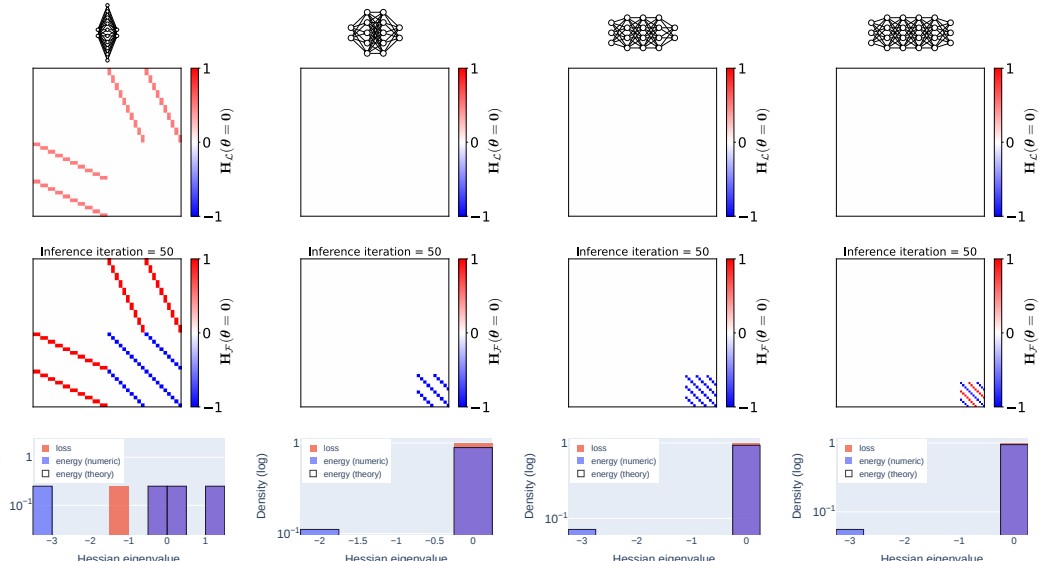

Figure 3: **Empirical verification of the Hessian at the origin of the equilibrated energy for DLNs tested on toy data.** We show the Hessian and its eigenspectrum at the origin of the MSE loss (*top*) and equilibrated energy (*middle*) for DLNs with Gaussian target $\mathbf{y} = -\mathbf{x}$ where $\mathbf{x} \sim \mathcal{N}(1, 0.1)$ (see §A.4 for details). Note that purple bars show overlapping loss and energy Hessian eigendensity. In the right panel, we vary one of the output dimensions to be $y_2 = x_2$. We confirm the strictness of the origin saddle in the equilibrated energy and observe an excellent numerical validation of our theoretical Hessian (Eq. 8). Figure 8 shows the same results for one-dimensional networks, and Figure 4 shows similar results for more realistic datasets.

observe that, when initialised close to the origin saddle, SGD takes increasingly more time to escape from the loss than the energy as a function of depth (for the same learning rate). Now we state the result more formally. The Hessian at the origin of the equilibrated energy turns out to be (see §A.3.3 for derivation)

$$\mathbf{H}_{\mathcal{F}^*}(\boldsymbol{\theta} = \mathbf{0}) = \begin{cases} \begin{bmatrix} \mathbf{0} & -\widetilde{\boldsymbol{\Sigma}}_{\mathbf{xy}} \otimes \mathbf{I}_{n_1} \\ -\mathbf{I}_{n_1} \otimes \widetilde{\boldsymbol{\Sigma}}_{\mathbf{yx}} & -\widetilde{\boldsymbol{\Sigma}}_{\mathbf{yy}} \otimes I_{n_{L-1}} \end{bmatrix}, & H = 1 \\[2em] \begin{bmatrix} \mathbf{0} & \dots & \mathbf{0} \\ \vdots & \ddots & \vdots \\ \mathbf{0} & \dots & -\widetilde{\boldsymbol{\Sigma}}_{\mathbf{yy}} \otimes I_{n_{L-1}} \end{bmatrix}, & H > 1 \end{cases}, \tag{8}$$

where $\widetilde{\boldsymbol{\Sigma}}_{\mathbf{yy}} := \frac{1}{N} \sum_i^N \mathbf{y}_i \mathbf{y}_i^T$ is the empirical output covariance. We see that, in contrast to the loss Hessian (Eq. 6), the energy Hessian has a non-zero last diagonal block given by $\partial^2 \mathcal{F}^* / \partial \mathbf{W}_L^2$, for any number of hidden layers $H$. It is then straightforward to show that the energy Hessian has always negative eigenvalues, since the output covariance is positive definite.

**Theorem 2** (Strictness of origin saddle of the equilibrated energy)**.** *The Hessian at the origin of the equilibrated energy (Eq. 5) for any DLN has at least one negative eigenvalue (see §A.3.3 for proof)*

$$\lambda_{min}(\mathbf{H}_{\mathcal{F}^*}(\boldsymbol{\theta} = \mathbf{0})) < 0, \quad \forall H \geq 1 \quad \text{[strict saddle, Def. 1]}. \tag{9}$$

Figures 3 & 4 show a perfect match between the theoretical (Eq. 8) and numerical Hessian at the origin of the equilibrated energy, which we computed for a range of DLNs on a random batch of toy as well as more realistic datasets.

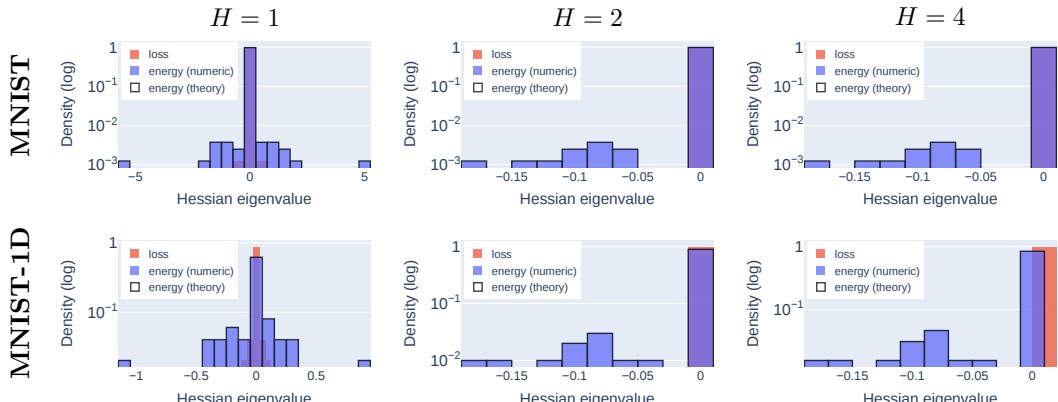

Figure 4: **Empirical verification of the Hessian eigenspectrum at the origin of the equilibrated energy for DLNs tested on more realistic datasets.** This shows similar results to Figure 3 for the more realistic datasets MNIST and MNIST-1D [16] (see §A.4 for details). We again find a perfect match between theory and experiment for DLNs with different number of hidden layers $H \in \{1, 2, 4\}$, confirming the strictness of the origin saddle of the equilibrated energy.

Theorem 2 proves that the origin is a strict saddle of the equilibrated energy for DLNs of any depth. This is in stark contrast to the MSE loss where it is only true for one-hidden-layer networks $H = 1$ (Eq. 7). The result predicts that, near the origin, (S)GD should escape the saddle faster on the equilibrated energy than on the loss given the same learning rate, and increasingly so as a function of depth. Figure 2 confirms this prediction for some toy linear networks, and Figures 5 & 6 in §4 clearly show that it holds for non-linear networks as well.

### 3.3 Analysis of other saddles

Is the origin a special case where the equilibrated energy has an easier-to-escape saddle than the loss? Or is this result pointing to something more general? Here we consider a specific type of non-strict saddle of the loss (of which the origin is one) and show that indeed they also become strict in the equilibrated energy. We address other saddle types experimentally in §4 and leave their theoretical study for future work.

Specifically, we consider saddles of rank zero, which for the MSE can be identified as critical points where the product of weight matrices is zero $\mathbf{W}_{L:1} = \mathbf{0}$ [1]. For the equilibrated energy (Eq. 5), we consider the critical points $\boldsymbol{\theta}^*(\mathbf{W}_L = \mathbf{0}, \mathbf{W}_{L-1:1} = \mathbf{0})$, since the last weight matrix needs to be null in order for the energy gradient to be zero (see §A.3.3 for an explanation). It turns out that at these critical points there exists a direction of negative curvature.

**Theorem 3** (Strictness of zero-rank saddles of the equilibrated energy). *Consider the set of critical points of the equilibrated energy (Eq. 5) $\boldsymbol{\theta}^*(\mathbf{W}_L = \mathbf{0}, \mathbf{W}_{L-1:1} = \mathbf{0})$ where $\mathbf{g}_{\mathcal{F}^*}(\boldsymbol{\theta}^*) = \mathbf{0}$. The Hessian at these points has at least one negative eigenvalue (see §A.3.6 for proof)*

$$\exists \lambda(\mathbf{H}_{\mathcal{F}^*}(\boldsymbol{\theta}^*)) < 0 \quad \text{[strict saddles, Def. 1]}. \tag{10}$$

Note that Theorem 2 can now be seen as a corollary of Theorem 3, although for the origin we derived the full Hessian. This result also stands in contrast to the (MSE) loss, where many of the considered critical points (specifically when 3 or more weight matrices are zero) are non-strict saddles as proved by [1]. The prediction is again that, in the vicinity of any of these saddles, PC should escape faster than BP with (S)GD given the same learning rate. For space reasons, the subsequent experiments focus only the origin as an example of a saddle covered by Theorem 3 (and Theorem 2), but §A.5 includes an empirical validation of another (zero-rank) strict saddle of the equilibrated energy (Figures 9, 10 & 12). Our code also makes it relatively easy to test for other saddles.

# 4 Experiments

Here we report experiments on linear and non-linear networks supporting our theoretical results as well as more general conjecture that all the saddles of the equilibrated energy are strict. In all the experiments, we trained networks with BP and PC using (S)GD with the same learning rate, since the goal is to test our theory of the saddle geometry of the equilibrated energy landscape. Code to reproduce all the results is available at https://github.com/francesco-innocenti/pc-saddles.

First, we compared the training loss (MSE) dynamics of linear and non-linear networks, including convolutional architectures, on standard image classification tasks with SGD initialised close to the origin (see §A.4 for details). For computational reasons, we did not run the BP-trained networks to convergence, underscoring the point that the origin saddle of the loss is highly degenerate and particularly hard to escape for first-order methods like SGD. In all cases, we observe that PC escapes the origin saddle substantially faster than BP (Figure 5), and Figure 11 shows that PC exhibits no vanishing gradients. We find practically the same results when initialising close to another non-strict saddle of the loss covered by Theorem 3 (Figure 12). These findings support our theoretical results beyond the linear case.

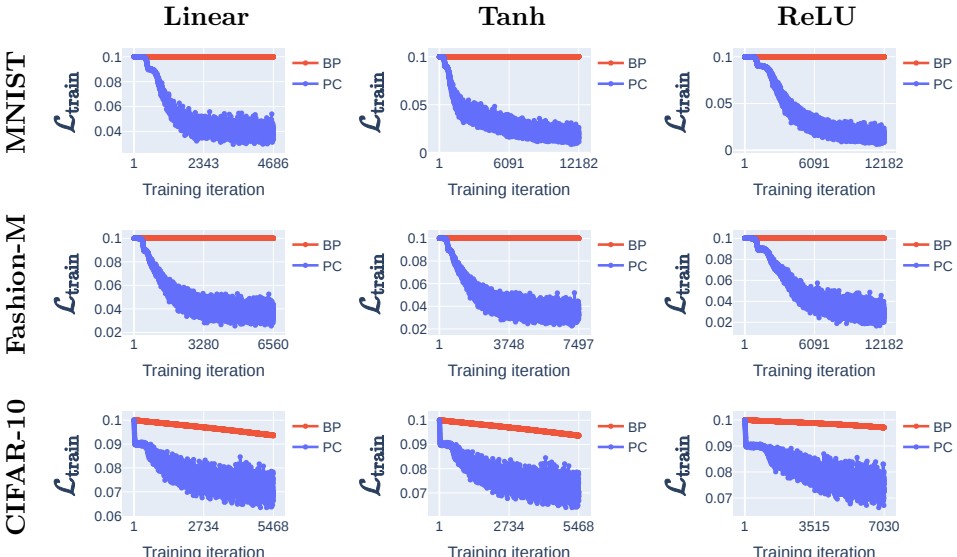

Figure 5: **PC escapes the origin saddle much faster than BP with SGD on non-linear networks.** We plot the training loss (MSE) for a representative run of BP and PC for linear and non-linear networks trained on standard image classification tasks (see §A.4 for details). All networks were initialised close to the origin with scale $\sigma = 5e^{-3}$), and trained with SGD and learning rate $\eta = 1e^{-3}$. The networks trained on MNIST and Fashion-MNIST had 5 fully connected layers, while those trained on CIFAR-10 had a convolutional architecture. See Figure 11 for the corresponding weight gradient norms during training. Results were consistent across different random seeds.

From Figure 5, we also observe a second plateau in the loss dynamics of PCNs, suggesting a saddle of higher rank (presumably rank 1). This is consistent with the saddle-to-saddle dynamics described for DLNs by [19], where for small initialisation GD transitions through a sequence of saddles, each representing a solution of increasing rank.

To explicitly test for higher-rank, non-strict saddles of the loss that we did not study theoretically, we replicated one of the experiments by [19, cf. Figure 1] on a matrix completion task. In particular, networks were trained to fit a rank-3 matrix, which meant that starting near origin GD visited 3 saddles (of successive rank 0, 1 and 2) before converging to a rank-3 solution as shown in Figure 6. We find that, when initialised near any of the saddles visited by BP, PC escapes quickly and does not show vanishing gradients (Figure 6), supporting the conjecture that all the saddles of the equilibrated energy are strict.

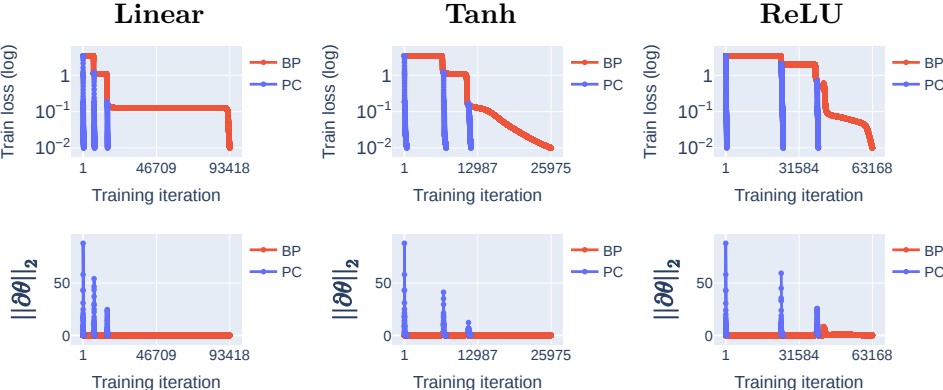

Figure 6: **PC quickly escapes higher-rank saddles visited by BP with GD on a matrix completion task.** We plot the training loss (*top*) and corresponding weight gradient norms of the loss (BP) and equilibrated energy (PC) (*bottom*) for networks ($H = 3$, $n_\ell = 100$) trained with full-batch GD to fit a random rank-3 matrix as studied by [19]. BP-trained networks were initialised near the origin with scale $\sigma = 5e^{-3}$, while PCNs were initialised at each saddle visited by BP (see §A.4 for details). Results were consistent across different random seeds.

## 5 Discussion

In summary, we took a first step in characterising the effective landscape on which PC learns—the energy landscape at the inference equilibrium. For DLNs, we first showed that the equilibrated energy is equal to a rescaled MSE loss with a weight-dependent rescaling (Theorem 1). This result corrects a previous mistake in the literature that the MSE loss is equal to the output energy [34] and that the total energy (Eq. 2) can therefore be decomposed into the loss and the other (hidden) energies (a relationship that only holds at the feedforward activity values). As we expand on below, Eq. 5 also enables further studies of the PC learning landscape.

We then proved that many non-strict saddle points of the MSE loss, specifically zero-rank saddles, become strict in the equilibrated energy of any DLN (Theorems 2 & 3). These saddles include the origin, making PC effectively more robust to vanishing gradients (Figures 6 & 11). We thoroughly validated our theory with experiments on both linear and non-linear architectures, and provided empirical support for the strictness of higher-rank saddles of the equilibrated energy. Based on these results, we conjecture that all the saddles of the equilibrated energy are strict. Overall, the PC inference process can therefore be interpreted as making the loss landscape more benign.

### 5.1 Implications

Our work goes significantly beyond existing theories of PC in terms of both explanatory and predictive power. Most previous works make non-standard assumptions or loose approximations that result in non-specific experimental predictions. For example, the interpretation of PC as implicit GD by [3] holds only for small batch sizes and specific layerwise rescalings of the activities and parameter learning rates. ([2] generalised this result to remove the activity rescalings but not the learning rate ones.) By contrast, linearity is the only major assumption made our theory, and we empirically verify that all the results hold for non-linear networks. Similarly, both [2] and [18] make second-order approximations of the energy to argue that PC makes use of Hessian information. However, our results clearly show that PC can leverage much higher-order information, turning highly degenerate, $H$-order saddles into strict (first-order) ones.

Previous theories have also struggled to explain why faster learning convergence with PC is not always observed depending on the task, model and optimiser [3, 54]. Our landscape analysis, while incomplete (more on this below), acknowledges these factors and their interplay, helping to explain inconsistent findings and predict when speed-ups can and cannot be expected. All things being equal, PC should converge faster on deep and *narrow* networks (though perhaps not too deep as we discuss below), since the distance between the origin saddle and standard initialisations scales with the network width [39]. This likely explains the speed-up reported by [54] on a narrow ($n_\ell = 64$)

15-layer fully connected network. However, in practice all things are not equal, and everything from not reaching an inference equilibrium to different datasets, architectures and optimisers all interact to determine convergence. This raises the question of whether minimising the equilibrated energy could be faster than the loss or lead to better performance, which we return to below.

More broadly, our landscape theory closely relates to the work of [56], who showed that learning in linear physical systems with equilibrium propagation [49, 50] has beneficial effects on the activity (rather than weight) Hessian. Studying these connections—and more generally the benefits of inference for learning in energy-based systems—could be an interesting future direction.

Our work has also implications for theories of credit assignment in the brain. In particular, our results put the recent principle of prospective configuration [54] for energy-based learning on a more solid theoretical footing, showing that PC inference can indeed facilitate learning by using high-order information. At the same time, they suggest that the claim of universally faster learning convergence with PC may have been overstated [54].

## 5.2 Limitations

We conclude by addressing the main limitations of our work. First, the strictness of the energy saddles we studied holds, by derivation, only at the exact inference equilibrium. We note that one does not need to reach equilibrium to improve the degeneracy of the loss saddles, and in this sense PC could be seen as a resource. However, in practice PC inference requires increasingly more iterations to converge on deeper networks, which aligns with our landscape theory since the loss saddles become more and more degenerate with depth. Our results therefore highlight the fundamental challenge of speeding up PC inference on deeper models if its benefits for learning are to be realised on large-scale tasks [40].

Even if this challenge is overcome, there seem to be two interlinked questions that ultimately matter for the practical training of deep networks. First, are there conditions under which the equilibrated energy can be minimised faster than the loss in a more compute- or memory-efficient manner, with at least equal performance? Optimisation tools such as Adam [24] and skip connections [17], for example, help to deal with the origin saddle at an increased memory cost. Could this trade off with the compute cost of PC inference? Characterising the inference cost of PC more formally would be a useful step in this direction.

Second, could there be scenarios where PC is slower or less efficient but at the benefit of significantly better performance? This is a hard question to address since we are far from having a theory of generalisation in deep learning [63, 20]. Given our origin saddle result (Theorem 2), however, it is interesting to note that on problems where a low-rank bias is useful (e.g. matrix completion, Figure 6), GD with small initialisations can converge to better-generalising solutions compared to standard initialisation [19].

Finally, understanding the overall convergence behaviour of PC would also require characterising other critical points of the equilibrated energy, especially its minima [14]. Our work, and Eq. 5 in particular, enables this. In §A.3.7, we present a preliminary investigation showing that, for linear chains, the global minima of the equilibrated energy are *flatter* than those of the MSE loss. This result potentially explains the common observation that PC convergence tends to slow down towards the end of training, but we leave its full implications for future work.

## Acknowledgements

F. I. is funded by the Sussex Neuroscience 4-year PhD Programme. E. M. A. acknowledges funding by the Deutsche Forschungsgemeinschaft (DFG, German Research Foundation) - Project number 442047500 through the Collaborative Research Center "Sparsity and Singular Structures" (SFB 1481). R. S. was supported by the Leverhulme Trust through the be.AI Doctoral Scholarship Programme in biomimetic embodied AI. C. L. B. was partially supported by the European Innovation Council (EIC) Pathfinder Challenges, Project METATOOL with Grant Agreement (ID: 101070940).

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

# A Appendix

**Contents**

## A.1 General notation and definitions

Matrices, vectors and scalars are denoted with bold capitals $\mathbf{A}$, bold lower-case characters $\mathbf{v}$ and non-bold characters $u$ or $U$, respectively. All vectors are by default column vectors $[\cdot] \in \mathbb{R}^{n \times 1}$, and $\mathrm{vec}_r(\cdot)$ denotes the row-wise vec operator. Following [52], unless otherwise stated we define matrix-by-matrix derivatives by row-vectorisation, using the numerator or Jacobian layout

$$\left(\frac{\partial \mathbf{A}}{\partial \mathbf{B}}\right)_{ij} := \frac{[\partial \, \mathrm{vec}_r(\mathbf{A})]_i}{[\partial \, \mathrm{vec}_r(\mathbf{B})^T]_j}, \tag{11}$$

such that the result is a matrix rather than a 4D tensor. Following from this, we will also use the rules

$$\frac{\partial \mathbf{ABC}}{\partial \mathbf{B}} = \mathbf{A} \otimes \mathbf{C}^T \tag{12}$$

$$\frac{\partial \mathbf{AB}}{\partial \mathbf{A}} = \mathbf{I}_m \otimes \mathbf{B}^T, \quad \mathbf{A} \in \mathbb{R}^{m \times n}, \mathbf{B} \in \mathbb{R}^{n \times p}. \tag{13}$$

## A.2 Related work

### A.2.1 Theories of predictive coding

**PC and BP.** [60] where the first to show that PC can approximate BP on multi-layer perceptrons when the influence of the input is upweighted relative to that of the output. [36] generalised this result to arbitrary computational graphs including convolutional and recurrent neural networks under the so-called "fixed prediction assumption". A variation of PC where weights are updated at precisely timed inference steps was later shown to compute exactly the same gradients as BP on multi-layer perceptrons [53], a result further generalised by [46] and [42]. [33] unified these and other approximation results from an energy-based modelling perspective. [62] proved that the time complexity of all of these PC variants is lower-bounded by BP.

**PC and other algorithms.** [13] provided an in-depth dynamical systems analysis of the inference convergence for PC variants approximating BP. [34] showed that for linear networks the PC inference equilibrium can be interpreted as an average of BP's feedforward pass values and the local targets computed by target propagation [30]. [54] proposed that PC and other energy-based algorithms implement a fundamentally different principle of credit assignment called "prospective configuration",

in that neurons first change their activity to align with the target and then update their weights to consolidate that activity pattern. For mini-batches of size one, [3] proved that PC approximates implicit gradient descent under specific layer-wise rescalings of the activities and parameter learning rates. [2] further showed that when this approximation holds, PC can be sensitive to Hessian information. Similarly, recent work cast PC as a second-order trust-region method [18].

### A.2.2 Saddle points and neural networks

Here we review some relevant theoretical and empirical work on (i) saddle points in the loss landscape of neural networks and (ii) the behaviour of different learning algorithms, especially (S)GD, near saddles. For more general reviews on the loss landscape and optimisation of neural networks, see [57] and [58].

**Saddles in the neural loss landscape**. This work began with [5] showing that for linear networks with one hidden layer, all critical points of the MSE loss are either global minima or strict saddle points (Def. 1). For the same model, [48] later showed saddle-to-saddle learning transitions for small initialisation and characterised the GD dynamics under specific assumptions on the data. [11] highlighted the prevalence of saddles, relative to local minima, in the high-dimensional non-convex loss of neural networks. In particular, they empirically demonstrated a qualitative similarity between the landscape of networks and random Gaussian error functions, where the higher the error a critical point is associated with, the more exponentially likely it is to be a saddle [8].

[23] famously generalised the [5] result that all local minima are global to arbitrarily deep linear networks (DLNs) under some weak assumptions on the data. This was simplified as well as extended under less strict assumptions by [29]. Importantly, [23] was the first to show that for neural networks with one hidden layer $H = 1$ all saddle points are strict (or first-order), while deeper networks have non-strict ($H$-order) saddles (for example at the origin where all the weights are zero). Several variations and extensions of this set of results have since been formulated [61, 64, 25, 65, 37, 66]. For our purposes, one important extension was made by [1], who characterised all the critical points of the MSE loss for DLNs to second-order, including strict and non-strict saddles.

**Learning near saddles**. This work can be traced to [15] who showed that SGD with added noise can converge in polynomial time on strict saddle functions. [27] proved a similar result that GD without any noise asymptotically escapes strict saddles for almost all initialisations. This was later generalised to other first-order methods [26]. [21] proved that another noisy version of GD converges with high probability to a second-order critical point in poly-logarithmic time depending on the dimension. For a review of these and other convergence results of GD and its variants, see [22]. [4] showed (i) that a further GD variant can be proved to converge to a third-order critical point and escape second-order saddles but at a high computational cost and (ii) that finding higher-order critical points is NP-hard.

[12] proved the important result that while standard GD with common initialisations will eventually escape strict saddles, it can take an exponential time to do so. This is in contrast to the perturbed GD versions mentioned above, which converge in polynomial time. Similarly, [51] proved that for linear chains or one-dimensional networks with unit width, the convergence time of GD scales exponentially with the depth. [39] analysed similar models and showed that both the gradients and curvature vanish with network depth unless the width is appropriately scaled. [39] suggested that this in part explains the success of adaptive gradient optimisers like Adam [24] which can adapt to flat curvature. Similarly, [55] showed that adaptive methods can escape saddle points faster by rescaling the gradient noise near critical points to be isotropic.

[19] conjectured a saddle-to-saddle dynamic where GD visits a sequence of saddles of increasing rank before converging to a sparse global minimum. A few works have also shown that in practice SGD can converge to second-order critical points that are non-strict saddles rather than minima [47, 7].

### A.3 Proofs and derivations

### A.3.1 Loss Hessian for DLNs

Here we derive the Hessian of the MSE loss (Eq. 1) with respect to the weights of arbitrary DLNs (§2.1); this is essentially a re-derivation of [52] with slightly different notation.[2] We then show how

---

[2]In particular, unlike [52] we make transposes of weight matrix products explicit.

the Hessian and its eigenspectrum at the origin ($\boldsymbol{\theta} = \mathbf{0}$) changes as a function of the number of hidden layers $H$. We start from the gradient of the loss for a given weight matrix

$$\frac{\partial \mathcal{L}}{\partial \mathbf{W}_\ell} = (\mathbf{W}_{L:\ell+1})^T (\mathbf{W}_{L:1} \mathbf{x} - \mathbf{y})(\mathbf{W}_{\ell-1:1} \mathbf{x})^T \tag{14}$$

$$= (\mathbf{W}_{L:\ell+1})^T (\mathbf{W}_{L:1} \widetilde{\boldsymbol{\Sigma}}_{\mathbf{xx}} - \widetilde{\boldsymbol{\Sigma}}_{\mathbf{yx}})(\mathbf{W}_{\ell-1:1})^T \in \mathbb{R}^{n_\ell \times n_{\ell-1}}, \tag{15}$$

where following previous works we take the empirical mean over the data matrices $\widetilde{\boldsymbol{\Sigma}}_{\mathbf{xx}} := \frac{1}{N} \sum_i^N \mathbf{x}_i \mathbf{x}_i^T$ and $\widetilde{\boldsymbol{\Sigma}}_{\mathbf{yx}} := \frac{1}{N} \sum_i^N \mathbf{y}_i \mathbf{x}_i^T$. For networks with at least one hidden layer, the origin is a critical point since the gradient is zero $\mathbf{g}_{\mathcal{L}}(\boldsymbol{\theta} = \mathbf{0}) = \mathbf{0}$. To characterise this point to second order, we look at the Hessian. Starting with the diagonal blocks of size $(n_\ell n_{\ell-1}) \times (n_\ell n_{\ell-1})$,

$$\frac{\partial^2 \mathcal{L}}{\partial \mathbf{W}_\ell^2} = (\mathbf{W}_{L:\ell+1})^T \mathbf{W}_{L:\ell+1} \otimes \mathbf{W}_{\ell-1:1} \widetilde{\boldsymbol{\Sigma}}_{\mathbf{xx}} (\mathbf{W}_{\ell-1:1})^T, \tag{16}$$

it is straightforward to see that at the origin this term collapses to the null matrix for any $l$.[3] To compute the $(n_k n_{k-1}) \times (n_\ell n_{\ell-1})$ off-diagonal blocks, we follow [52] and write the distinct contributions as follows

$$\forall k \neq \ell, \quad \widetilde{\mathbf{H}}_{\mathcal{L}} := \frac{\partial^2 \mathcal{L}}{\partial \mathbf{W}_k \partial \mathbf{W}_\ell} = (\mathbf{W}_{L:\ell+1})^T \mathbf{W}_{L:k+1} \otimes \mathbf{W}_{\ell-1:1} \widetilde{\boldsymbol{\Sigma}}_{\mathbf{xx}} (\mathbf{W}_{k-1:1})^T \tag{17}$$

$$\forall k > \ell, \quad \widehat{\mathbf{H}}_{\mathcal{L}} := \frac{\partial^2 \mathcal{L}}{\partial \mathbf{W}_k^T \partial \mathbf{W}_\ell} = (\mathbf{W}_{k-1:\ell+1})^T \otimes \mathbf{W}_{\ell-1:1}(\mathbf{W}_{L:1}\widetilde{\boldsymbol{\Sigma}}_{\mathbf{xx}} - \widetilde{\boldsymbol{\Sigma}}_{\mathbf{yx}})^T \mathbf{W}_{L:k+1} \tag{18}$$

$$\forall k < \ell, \quad \widehat{\mathbf{H}}_{\mathcal{L}} := \frac{\partial^2 \mathcal{L}}{\partial \mathbf{W}_k^T \partial \mathbf{W}_\ell} = (\mathbf{W}_{L:\ell+1})^T (\mathbf{W}_{L:1}\widetilde{\boldsymbol{\Sigma}}_{\mathbf{xx}} - \widetilde{\boldsymbol{\Sigma}}_{\mathbf{yx}})(\mathbf{W}_{k-1:1})^T \otimes (\mathbf{W}_{\ell-1:k+1})^T. \tag{19}$$

At the origin, these blocks always vanish except for networks with one hidden layer, where as shown by [48] they are characterised by the empirical input-output covariance, e.g. for $k < \ell, \partial^2 \mathcal{L}/\partial \mathbf{W}_k \partial \mathbf{W}_\ell(\boldsymbol{\theta} = \mathbf{0}) = -\widetilde{\boldsymbol{\Sigma}}_{\mathbf{xy}} \otimes \mathbf{I}_n, H = 1$. Putting the above facts together, we can now write the full loss Hessian at the origin for different number of hidden layers:

$$\mathbf{H}_{\mathcal{L}}(\boldsymbol{\theta} = \mathbf{0}) = \begin{cases} \begin{bmatrix} \mathbf{0} & -\widetilde{\boldsymbol{\Sigma}}_{\mathbf{xy}} \otimes \mathbf{I}_{n_1} \\ -\mathbf{I}_{n_1} \otimes \widetilde{\boldsymbol{\Sigma}}_{\mathbf{yx}} & \mathbf{0} \end{bmatrix}, & H = 1 \quad \text{[strict saddle]} \\ \\ \begin{bmatrix} \mathbf{0} & \ldots & \mathbf{0} \\ \vdots & \ddots & \vdots \\ \mathbf{0} & \ldots & \mathbf{0} \end{bmatrix} = \mathbf{0}_p, & H > 1 \quad \text{[non-strict saddle]} \end{cases} \tag{20}$$

For one-hidden-layer networks, the Hessian is indefinite, with positive and negative eigenvalues given by the empirical input-output covariance, as described by [48]. For any DLN with more than one hidden layer, the Hessian is zero, and the origin is therefore a second-order critical point. In the general case, this point is a non-strict saddle because some higher-order derivative of the loss depending on the network depth will contain at least one negative escape direction. More specifically, for a network with $L$ layers, all the $L - 1$ derivatives vanish, and negative directions will be found in the derivatives of order $\geq L$.

### A.3.2  Equilibrated energy for DLNs

Here we derive an exact solution to the PC energy (Eq. 2) of DLNs at the inference equilibrium (Theorem 1, Eq. 5), $\mathcal{F}|_{\partial \mathcal{F}/\partial \mathbf{z}=0}$, which we will abbreviate as $\mathcal{F}^*$. This turns out to be a non-trivial rescaled MSE loss where the rescaling depends on covariances of the network weight matrices. To highlight the difference with the loss, recall that the standard MSE (Eq. 1) for a DLN implicitly defines the following generative model

$$\mathbf{y} \sim \mathcal{N}(\mathbf{W}_{L:1}\mathbf{x}, \boldsymbol{\Sigma}) \tag{21}$$

---

[3]To be precise, this is true for any network with at least one hidder layer $H \geq 1$. For zero-hidden-layer networks $H = 0$—which are equivalent to a linear regression problem—the origin is a not a critical point, $\mathbf{g}_{\mathcal{L}}(\boldsymbol{\theta} = \mathbf{0}) = -\widetilde{\boldsymbol{\Sigma}}_{\mathbf{yx}}$, and the Hessian is constant $\mathbf{H}_{\mathcal{L}} = \mathbf{I}_{d_y} \otimes \widetilde{\boldsymbol{\Sigma}}_{\mathbf{xx}}$.

where the target is modelled as a Gaussian with a mean given by the network function and some covariance $\mathbf{\Sigma}$. In a PC network, by contrast, the activity of *each hidden layer*–and not just the output–is modelled as a Gaussian (see §2.2)

$$\mathbf{z}_\ell \sim \mathcal{N}(\mathbf{W}_\ell \mathbf{z}_{\ell-1}, \mathbf{I}_\ell), \tag{22}$$

where $\mathbf{z}_0 := \mathbf{x}$ and $\mathbf{z}_L := \mathbf{y}$. Now, to work out the generative model for the target implied by this hierarchical Gaussian model, we can simply "unfold" the model at each layer. Specifically, we can reparameterise the activity of each hidden layer as a noisy function of the previous layer and so on recursively up to the first layer

$$\mathbf{z}_1 = \mathbf{W}_1 \mathbf{z}_0 + \boldsymbol{\xi}_1 \tag{23}$$
$$\mathbf{z}_2 = \mathbf{W}_2 \mathbf{z}_1 + \boldsymbol{\xi}_2 = \mathbf{W}_2 \mathbf{W}_1 \mathbf{x} + \mathbf{W}_2 \boldsymbol{\xi}_1 + \boldsymbol{\xi}_2 \tag{24}$$
$$\mathbf{z}_3 = \mathbf{W}_3 \mathbf{z}_2 + \boldsymbol{\xi}_3 = \mathbf{W}_3 \mathbf{W}_2 \mathbf{W}_1 \mathbf{x} + \mathbf{W}_3 \mathbf{W}_2 \boldsymbol{\xi}_1 + \mathbf{W}_3 \boldsymbol{\xi}_2 + \boldsymbol{\xi}_3 \tag{25}$$
$$\vdots$$

where $\boldsymbol{\xi}_\ell \sim \mathcal{N}(\mathbf{0}, \mathbf{I}_\ell)$ is white Gaussian noise. The last layer can then be written as

$$\mathbf{z}_L = \mathbf{W}_L \mathbf{z}_{L-1} + \boldsymbol{\xi}_L \tag{26}$$

$$= \mathbf{W}_{L:1} \mathbf{z}_0 + \sum_{\ell=2}^{L} \mathbf{W}_{L:\ell} \boldsymbol{\xi}_{\ell-1} + \boldsymbol{\xi}_L. \tag{27}$$

We can now derive the implicit generative model for the target by taking the expectation and variance of Eq. 27:

$$\mathbf{y} \sim \mathcal{N}\left( \mathbf{W}_{L:1} \mathbf{x}, \mathbf{I}_L + \sum_{\ell=2}^{L} (\mathbf{W}_{L:\ell})(\mathbf{W}_{L:\ell})^T \right). \tag{28}$$

We therefore observe that, in contrast to the loss (Eq. 21), PC implicitly models the target with a non-identity covariance depending on a chained covariance of the previous layers which in turns depends only on the weights. It follows that, at the exact inference equilibrium where that implicit generative model holds, the PC energy is simply the following rescaled MSE loss

$$\mathcal{F}^* = \frac{1}{2N} \sum_i^N (\mathbf{y}_i - \mathbf{W}_{L:1} \mathbf{x}_i)^T \mathbf{S}^{-1} (\mathbf{y}_i - \mathbf{W}_{L:1} \mathbf{x}_i) \tag{29}$$

where the rescaling is $\mathbf{S} = \mathbf{I}_{d_y} + \sum_{\ell=2}^{L} (\mathbf{W}_{L:\ell})(\mathbf{W}_{L:\ell})^T$. One can also arrive at this expression by explicitly solving for the activities $\partial \mathcal{F}/\partial \mathbf{z} = 0$ and plugging the solution back into the energy, although the calculation becomes much more involved. Note that a generative model with non-identity covariances at each layer would lead to a different rescaling, but we do not consider this here to remain as close as possible to what is done in practice.

### A.3.3 Hessian of the equilibrated energy for DLNs

Here we derive the Hessian at the origin of the equilibrated energy for DLNs, following the calculation of the loss Hessian (§A.3.1). Section A.3.5 shows an equivalent derivation for one-dimensional linear networks, which preserves all the key the intuitions and is easier to follow. We start from the equilibrated energy we derived previously for DLNs (§A.3.2, Eq. 29), which turned out to be the following rescaled MSE loss

$$\mathcal{F}^* = \frac{1}{2N} \sum_i^N \mathbf{r}_i^T \mathbf{S}^{-1} \mathbf{r}_i \tag{30}$$

where $\mathbf{S} = \mathbf{I}_{d_y} + \sum_{\ell=2}^{L} (\mathbf{W}_{L:\ell})(\mathbf{W}_{L:\ell})^T$, and we denote the residual error for a given data sample as $\mathbf{r}_i := (\mathbf{y}_i - \mathbf{W}_{L:1} \mathbf{x}_i)$. In the general case, both the residual and the rescaling depend on $\mathbf{W}_\ell$, so to take the gradient of the equilibrated energy we need the product rule. For simplicity, and similar to the characterisation of the off-diagonal blocks of the loss Hessian (§A.3.1), we write the two

contributions separately, as follows

$$\mathbf{A} := \frac{1}{2N} \sum_i^N \frac{\partial \mathbf{r}_i^T}{\partial \mathbf{W}_\ell} \mathbf{S}^{-1} \frac{\partial \mathbf{r}_i}{\partial \mathbf{W}_\ell} = (\mathbf{W}_{L:\ell+1})^T \mathbf{S}^{-1} (\mathbf{W}_{L:1} \widetilde{\mathbf{\Sigma}}_{\mathbf{xx}} - \widetilde{\mathbf{\Sigma}}_{\mathbf{yx}})(\mathbf{W}_{\ell-1:1})^T \quad (31)$$

$$\mathbf{B} := \frac{1}{2N} \sum_i^N \mathbf{r}_i^T \frac{\partial \mathbf{S}^{-1}}{\partial \mathbf{W}_\ell} \mathbf{r}_i = -\frac{1}{N} \sum_i^N \mathbf{S}^{-1} \mathbf{r}_i \mathbf{r}_i^T \mathbf{S}^{-1} \frac{\partial \mathbf{S}}{\partial \mathbf{W}_\ell}. \quad (32)$$

where in Eq. 32 $\partial \mathbf{S}/\partial \mathbf{W}_\ell$ is a 4D tensor, and we use the rule $\partial \mathbf{a}^T \mathbf{X} \mathbf{b}/\partial \mathbf{X} = -\mathbf{X}^{-T} \mathbf{a} \mathbf{b}^T \mathbf{X}^{-T}$. The first term $\mathbf{A}$ is simply a rescaled loss gradient, while the second term $\mathbf{B}$ depends on the derivative of the rescaling. Note that for $\mathbf{W}_1$ the gradient collapses to the first term since the rescaling does not depend on it, $\partial \mathcal{F}^*/\partial \mathbf{W}_1 = (\mathbf{W}_{L:2})^T \mathbf{S}^{-1}(\mathbf{W}_{L:1} \widetilde{\mathbf{\Sigma}}_{\mathbf{xx}} - \widetilde{\mathbf{\Sigma}}_{\mathbf{yx}})$.

As an aside relevant to the zero-rank saddles analysed in §3.3, we note that, in contrast to the loss, $\mathbf{W}_L = \mathbf{0}$ is a necessary (though not sufficient) condition for the energy gradient to be zero. This is because the derivative of the rescaling $\partial \mathbf{S}/\partial \mathbf{W}_\ell$ needs to be zero in order for the gradient term $\mathbf{B}$ to vanish, and it has one term linear in the last weight matrix.

As for the loss (§A.3.1), the origin is a critical point of the energy since $\mathbf{g}_{\mathcal{F}^*}(\boldsymbol{\theta} = \mathbf{0}) = \mathbf{0}$. For $\mathbf{B}$, this is because while the rescaling at zero is the identity, the derivative of the rescaling vanishes since it is linear with respect to any weight matrix:

$$\mathbf{S}^{-1}(\boldsymbol{\theta} = \mathbf{0}) = \mathbf{I}_{d_y} \quad (33)$$

$$\frac{\partial \mathbf{S}}{\partial \mathbf{W}_\ell}(\boldsymbol{\theta} = \mathbf{0}) = \mathbf{0}. \quad (34)$$

Calculating the Hessian involves multiple application of the product rule, so for simplicity we analyse the contribution of the derivative of each term (Eqs. 31 & 32) at the origin. Because the first term is simply a rescaling of the loss, and given Eq. 33, its second derivative at zero is always zero with respect to the same weight matrix,

$$k = \ell, \quad \frac{\partial \mathbf{A}}{\partial \mathbf{W}_k}(\boldsymbol{\theta} = \mathbf{0}) = \mathbf{0}, \quad H \geq 1. \quad (35)$$

As for the loss, this term is also zero with respect to some other weight matrix $k \neq \ell$ except for the case of a one-hidden-layer network

$$k \neq \ell, \quad \frac{\partial \mathbf{A}}{\partial \mathbf{W}_k}(\boldsymbol{\theta} = \mathbf{0}) = \begin{cases} -\mathbf{I}_{n_1} \otimes \widetilde{\mathbf{\Sigma}}_{\mathbf{yx}}, & k > \ell, H = 1 \\ -\widetilde{\mathbf{\Sigma}}_{\mathbf{xy}} \otimes \mathbf{I}_{n_1}, & k < \ell, H = 1 \\ \mathbf{0}, & H > 1 \end{cases} \quad (36)$$

The second derivative of $\mathbf{B}$ requires a 5-fold application of the product rule, involving the first derivative of the residual (and its transpose) and the first and second derivatives of the rescaling. As shown above (Eq. 34), the first derivative of the rescaling at the origin is zero, and the derivative of the residual with respect to any weight matrix at zero is always zero for any network with one or more hidden layers, $\partial \mathbf{r}/\partial \mathbf{W}_k(\boldsymbol{\theta} = \mathbf{0}) = \mathbf{0}, H \geq 1$. The second derivative of the rescaling, however, is non-zero for the special case of the last weight matrix with respect to itself:

$$\frac{\partial^2 \mathbf{S}}{\partial \mathbf{W}_k \partial \mathbf{W}_\ell}(\boldsymbol{\theta} = \mathbf{0}) = \begin{cases} \mathbf{I}_{n_{L-1}}, & \ell = k = L \\ \mathbf{0}, & \text{else} \end{cases}, \quad (37)$$

which means that at zero $\mathbf{B}$ takes the following form

$$\frac{\partial \mathbf{B}}{\partial \mathbf{W}_k}(\boldsymbol{\theta} = \mathbf{0}) = \begin{cases} -\widetilde{\mathbf{\Sigma}}_{\mathbf{yy}} \otimes I_{n_{L-1}}, & \ell = k = L \\ \mathbf{0}, & \text{else} \end{cases}, \quad (38)$$

where $\widetilde{\mathbf{\Sigma}}_{\mathbf{yy}} := \frac{1}{N}\sum_i^N \mathbf{y}_i\mathbf{y}_i^T$ is the empirical output covariance matrix. Drawing all these observations together, we can write the full Hessian at the origin of the equilibrated energy for different number of hidden layers:

$$\mathbf{H}_{\mathcal{F}^*}(\boldsymbol{\theta}=\mathbf{0}) = \begin{cases} \begin{bmatrix} \mathbf{0} & -\widetilde{\mathbf{\Sigma}}_{\mathbf{xy}}\otimes\mathbf{I}_{n_1} \\ -\mathbf{I}_{n_1}\otimes\widetilde{\mathbf{\Sigma}}_{\mathbf{yx}} & -\widetilde{\mathbf{\Sigma}}_{\mathbf{yy}}\otimes I_{n_{L-1}} \end{bmatrix}, & H=1 \quad \text{[strict saddle]} \\[2em] \begin{bmatrix} \mathbf{0} & \cdots & \mathbf{0} \\ \vdots & \ddots & \vdots \\ \mathbf{0} & \cdots & -\widetilde{\mathbf{\Sigma}}_{\mathbf{yy}}\otimes I_{n_{L-1}} \end{bmatrix}, & H>1 \quad \text{[strict saddle]} \end{cases} \qquad (39)$$

We see that, compared to the loss Hessian (Eq. 20), the energy Hessian has a non-zero last diagonal block given for any $H$. We note, but do not investigate in any depth, the potential connection with target propagation [30, 34]. The one-hidden-layer case is fully derived in the next section (§A.3.4). It is straightforward to show that these matrices have negative eigenvalues

$$H \geq 1, \quad \lambda_{\min}(\mathbf{H}_{\mathcal{F}^*}(\boldsymbol{\theta}=\mathbf{0})) < 0, \quad \forall y_i \neq 0 \qquad (40)$$

since $\mathbf{AA}^T$ is positive definite $\forall \mathbf{A} \neq \mathbf{0}$. The origin is therefore a strict saddle (Def. 1) of the equilibrated energy. This is in stark contrast to the MSE loss, which has a strict origin saddle only for one-hidden-layer networks and a non-strict saddle of order $H$ for any deeper network. For the general case $H > 1$, the negative curvature of the energy Hessian is given only by the output-output covariance $\widetilde{\mathbf{\Sigma}}_{\mathbf{yy}}$. This means that, in the vicinity of the origin saddle, GD steps of equal size on the equilibrated energy will escape the saddle faster (at a rate depending on the output structure) than on the loss, and increasingly so as a function of depth. In §4, we empirically verify this prediction experimentally on linear as well as non-linear architectures (including convolutional) trained on different datasets.

### A.3.4 Example: 1-hidden layer linear network

Here we show an example calculation comparing the Hessian at the origin of the loss and equilibrated energy for DLNs with a single hidden layer $H = 1$. For this case, the MSE loss and equilibrated energy are

$$\mathcal{L} = \frac{1}{2N}\sum_i^N ||\mathbf{y}_i - \mathbf{W}_2\mathbf{W}_1\mathbf{x}_i||^2 \qquad (41)$$

$$\mathcal{F}^* = \frac{1}{2N}\sum_i^N (\mathbf{y}_i - \mathbf{W}_2\mathbf{W}_1\mathbf{x}_i)^T(\mathbf{I}_{d_y} + \mathbf{W}_2\mathbf{W}_2^T)^{-1}(\mathbf{y}_i - \mathbf{W}_2\mathbf{W}_1\mathbf{x}_i) \qquad (42)$$

where $\mathbf{x} \in \mathbb{R}^{d_x}, \mathbf{y} \in \mathbb{R}^{d_y}, \mathbf{W}_1 \in \mathbb{R}^{n\times d_x}, \mathbf{W}_2 \in \mathbb{R}^{d_y\times n}$. We now show the weight gradients, first of the loss

$$\frac{\partial\mathcal{L}}{\partial\mathbf{W}_1} = \mathbf{W}_2^T\mathbf{W}_2\mathbf{W}_1\widetilde{\mathbf{\Sigma}}_{\mathbf{xx}} - \mathbf{W}_2^T\widetilde{\mathbf{\Sigma}}_{\mathbf{yx}} \qquad (43)$$

$$\frac{\partial\mathcal{L}}{\partial\mathbf{W}_2} = \mathbf{W}_2\mathbf{W}_1\widetilde{\mathbf{\Sigma}}_{\mathbf{xx}}\mathbf{W}_1^T - \widetilde{\mathbf{\Sigma}}_{\mathbf{yx}}\mathbf{W}_1^T, \qquad (44)$$

and then of the equilibrated energy

$$\frac{\partial\mathcal{F}^*}{\partial\mathbf{W}_1} = \mathbf{W}_2^T\mathbf{S}^{-1}\mathbf{W}_2\mathbf{W}_1\widetilde{\mathbf{\Sigma}}_{\mathbf{xx}} - \mathbf{W}_2^T\mathbf{S}^{-1}\widetilde{\mathbf{\Sigma}}_{\mathbf{yx}} \qquad (45)$$

$$\frac{\partial\mathcal{F}^*}{\partial\mathbf{W}_2} = \mathbf{S}^{-1}(\mathbf{W}_2\mathbf{W}_1\widetilde{\mathbf{\Sigma}}_{\mathbf{xx}} - \widetilde{\mathbf{\Sigma}}_{\mathbf{yx}})\mathbf{W}_1^T - \mathbf{S}^{-1}\mathbf{\Psi}\mathbf{S}^{-1}\mathbf{W}_2, \qquad (46)$$

where we denote the empirical mean of the residual as $\mathbf{\Psi} := \frac{1}{N}\sum_i^N \mathbf{r}_i\mathbf{r}_i^T$. The origin is a critical point of the both the loss and the equilibrated energy since $\mathbf{g}_{\mathcal{L}}(\boldsymbol{\theta}=\mathbf{0}) = \mathbf{g}_{\mathcal{F}^*}(\boldsymbol{\theta}=\mathbf{0}) = \mathbf{0}$. We now compute the Hessian blocks, expressing the off-diagonals at the origin for simplicity, again first for

the loss

$$\frac{\partial^2 \mathcal{L}}{\partial \mathbf{W}_1^2} = \mathbf{W}_2^T \mathbf{W}_2 \otimes \widetilde{\boldsymbol{\Sigma}}_{\mathbf{xx}} \tag{47}$$

$$\frac{\partial^2 \mathcal{L}}{\partial \mathbf{W}_2^2} = \mathbf{I}_{d_x} \otimes \mathbf{W}_1 \widetilde{\boldsymbol{\Sigma}}_{\mathbf{xx}} \mathbf{W}_1^T \tag{48}$$

$$\frac{\partial^2 \mathcal{L}}{\partial \mathbf{W}_2 \partial \mathbf{W}_1}(\boldsymbol{\theta} = \mathbf{0}) = -\mathbf{I}_n \otimes \widetilde{\boldsymbol{\Sigma}}_{\mathbf{yx}}, \tag{49}$$

and then for the energy

$$\frac{\partial^2 \mathcal{F}^*}{\partial \mathbf{W}_1^2} = \mathbf{W}_2^T \mathbf{S}^{-1} \mathbf{W}_2 \otimes \widetilde{\boldsymbol{\Sigma}}_{\mathbf{xx}} \tag{50}$$

$$\frac{\partial^2 \mathcal{F}^*}{\partial \mathbf{W}_2^2} = \mathbf{S}^{-1} \otimes \mathbf{W}_1 \widetilde{\boldsymbol{\Sigma}}_{\mathbf{xx}} \mathbf{W}_1^T - \mathbf{S}^{-1} \boldsymbol{\Psi} \mathbf{S}^{-1} \otimes \mathbf{I}_n \tag{51}$$

$$\frac{\partial^2 \mathcal{F}^*}{\partial \mathbf{W}_2 \partial \mathbf{W}_1}(\boldsymbol{\theta} = \mathbf{0}) = -\mathbf{I}_n \otimes \widetilde{\boldsymbol{\Sigma}}_{\mathbf{yx}}. \tag{52}$$

At the origin, the Hessians become

$$\mathbf{H}_{\mathcal{L}}(\boldsymbol{\theta} = \mathbf{0}) = \begin{bmatrix} \mathbf{0} & -\widetilde{\boldsymbol{\Sigma}}_{\mathbf{xy}} \otimes \mathbf{I}_n \\ -\mathbf{I}_n \otimes \widetilde{\boldsymbol{\Sigma}}_{\mathbf{yx}} & \mathbf{0} \end{bmatrix} \tag{53}$$

$$\mathbf{H}_{\mathcal{F}^*}(\boldsymbol{\theta} = \mathbf{0}) = \begin{bmatrix} \mathbf{0} & -\widetilde{\boldsymbol{\Sigma}}_{\mathbf{xy}} \otimes \mathbf{I}_n \\ -\mathbf{I}_n \otimes \widetilde{\boldsymbol{\Sigma}}_{\mathbf{yx}} & -\widetilde{\boldsymbol{\Sigma}}_{\mathbf{yy}} \otimes \mathbf{I}_n \end{bmatrix}. \tag{54}$$

### A.3.5 Hessian of the equilibrated energy for linear chains

Here we include a derivation the Hessian of the equilibrated energy (as well as its eigenstructure at the origin) for linear chains or networks of unit width $w_{L:1}x$ where $n_0 = \cdots = n_L = 1$. This follows the derivation for the wide case (§A.3.3), but it reveals all the key insights and is easier to follow. For the scalar case, the implicit generative model of the target defined by PC (see §A.3.2) is

$$y \sim \mathcal{N}\left(w_{L:1}x, 1 + \sum_{\ell=2}^{L}(w_{L:\ell})^2\right), \tag{55}$$

leading to the following rescaled loss

$$\mathcal{F}^* = \mathcal{L}/s, \quad s = 1 + \sum_{\ell=2}^{L}(w_{L:\ell})^2 \tag{56}$$

where $\mathcal{L} = \frac{1}{2N}\sum_{i}^{N}(y_i - w_{L:1}x_i)^2$. The weight gradient of the equilibrated energy is

$$\frac{\partial \mathcal{F}^*}{\partial w_i} = \begin{cases} \frac{1}{s}\frac{\partial \mathcal{L}}{\partial w_i}, & i = 1 \\ \frac{1}{s}\frac{\partial \mathcal{L}}{\partial w_i} - \frac{1}{s^2}\mathcal{L}\frac{\partial s}{\partial w_i}, & i > 1 \end{cases} \tag{57}$$

where the loss gradient is $\partial \mathcal{L}/\partial w_i = -w_{L:1 \neq i}xr$ with residual error $r = (y - w_{L:1}x)$. As shown in §A.3.2, The origin is a critical point of both the loss and the equilibrated energy since their gradients are zero $\mathbf{g}_{\mathcal{L}}(\boldsymbol{\theta} = \mathbf{0}) = \mathbf{0}, \mathbf{g}_{\mathcal{F}^*}(\boldsymbol{\theta} = \mathbf{0}) = \mathbf{0}$. We now show the Hessians, first of the loss

$$\frac{\partial^2 \mathcal{L}}{\partial w_i \partial w_j} = \begin{cases} (w_{L:1 \neq i})^2 x^2, & i = j \\ (w_{L:1 \neq i,j})(2w_{L:1}x^2 - xy), & i \neq j \end{cases}, \tag{58}$$

and then of the energy

$$
\frac{\partial^2 \mathcal{F}^*}{\partial w_i \partial w_j} =
\begin{cases}
\frac{1}{s} \frac{\partial^2 \mathcal{L}}{\partial w_i \partial w_j}, & i = j = 1 \\[2ex]
\frac{1}{s} \frac{\partial^2 \mathcal{L}}{\partial w_i \partial w_j} - \frac{1}{s^2} \frac{\partial \mathcal{L}}{\partial w_i} \frac{\partial s}{\partial w_j}, & i = 1, \quad j > 1 \\[2ex]
\frac{1}{s} \frac{\partial^2 \mathcal{L}}{\partial w_i \partial w_j} - \frac{1}{s^2} \frac{\partial \mathcal{L}}{\partial w_i} \frac{\partial s}{\partial w_j} + \frac{1}{s^2} \frac{\partial \mathcal{L}}{\partial w_j} \frac{\partial s}{\partial w_i} + \frac{1}{s^2} \mathcal{L} \frac{\partial^2 s}{\partial w_i \partial w_j} - \frac{2}{s^3} \frac{\partial s}{\partial w_j} \mathcal{L} \frac{\partial s}{\partial w_i}, & i, j > 1
\end{cases}
\tag{59}
$$

Generalising the one-hidden-unit case shown by [18], at the origin the Hessians reduce to

$$
\mathbf{H}_{\mathcal{L}}(\boldsymbol{\theta} = \mathbf{0}) =
\begin{cases}
\begin{bmatrix} 0 & -xy \\ -xy & 0 \end{bmatrix}, & H = 1 \quad \text{[strict saddle]} \\[3ex]
\begin{bmatrix} 0 & \cdots & 0 \\ \vdots & \ddots & \vdots \\ 0 & \cdots & 0 \end{bmatrix} = \mathbf{0}_p, & H > 1 \quad \text{[non-strict saddle]}
\end{cases}
\tag{60}
$$

$$
\mathbf{H}_{\mathcal{F}^*}(\boldsymbol{\theta} = \mathbf{0}) =
\begin{cases}
\begin{bmatrix} 0 & -xy \\ -xy & -y^2 \end{bmatrix}, & H = 1 \quad \text{[better-conditioned strict saddle]} \\[3ex]
\begin{bmatrix} 0 & \cdots & 0 \\ \vdots & \ddots & \vdots \\ 0 & \cdots & -y^2 \end{bmatrix}, & H > 1 \quad \text{[strict saddle]}
\end{cases}
\tag{61}
$$

For one-hidden-layer networks $H = 1$, the Hessian eigenvalues of the loss and energy are $\lambda(\mathbf{H}_{\mathcal{L}}(\boldsymbol{\theta} = \mathbf{0})) = \pm xy$, $\lambda(\mathbf{H}_{\mathcal{F}^*}(\boldsymbol{\theta} = \mathbf{0})) = (-y^2 \pm y\sqrt{4x^2 + y^2})/2$, respectively. In this case, the eigenvalues of the energy turn out to be smaller than those of the loss,

$$
H = 1, \quad \lambda(\mathbf{H}_{\mathcal{F}^*}(\boldsymbol{\theta} = \mathbf{0})) < \lambda(\mathbf{H}_{\mathcal{L}}(\boldsymbol{\theta} = \mathbf{0})), \quad \forall x, y \neq 0
\tag{62}
$$

following from the fact that the square root of a sum is smaller than the sum of the square roots, $\sqrt{a^2 + b^2} < \sqrt{a^2} + \sqrt{b^2}, \forall a, b \neq 0$. This means that, in this particular case, the strict saddle of the equilibrated energy is better conditioned (i.e. easier to escape) than that of the loss. For deeper networks, the Hessian of the loss is zero, and it is easy to see that that of the energy has zero eigenvalues of multiplicity $L - 1$ and a single negative eigenvalue given by the target squared

$$
H > 1, \quad \lambda_{\min}(\mathbf{H}_{\mathcal{F}^*}(\boldsymbol{\theta} = \mathbf{0})) = -y^2.
\tag{63}
$$

### A.3.6 Strictness of zero-rank saddles of the equilibrated energy

Here we prove the strictness of the zero-rank saddles of the equilibrated energy (Theorem 3). It is easy to check via Eqs. 31 & 32 that any point $\boldsymbol{\theta}^*$ such that $(\mathbf{W}_L = \mathbf{0}, \mathbf{W}_{L-1:1} = \mathbf{0})$ is a critical point. Now let's prove that the Hessian at $\boldsymbol{\theta}^*$ has a negative eigenvalue. To do this, we rely on the Taylor expansion of the function around $\boldsymbol{\theta}^*$. Since $\mathbf{g}_{\mathcal{F}^*}(\boldsymbol{\theta}^*) = \mathbf{0}$, we have for any $\hat{\boldsymbol{\theta}}$ and any $\delta \to 0$,

$$
\mathcal{F}^*(\boldsymbol{\theta}^* + \delta\hat{\boldsymbol{\theta}}) = \mathcal{F}^*(\boldsymbol{\theta}^*) + \frac{1}{2}\delta^2 \hat{\boldsymbol{\theta}}^T \mathbf{H}_{\mathcal{F}^*}(\boldsymbol{\theta}^*)\hat{\boldsymbol{\theta}} + \mathcal{O}(\delta^3).
\tag{64}
$$

Hence by unicity of the Taylor expansion, if we can find $\hat{\boldsymbol{\theta}}$ such that $\mathcal{F}^*(\boldsymbol{\theta}^* + \delta\hat{\boldsymbol{\theta}}) = \mathcal{F}^*(\boldsymbol{\theta}^*) - c\delta^2 + \mathcal{O}(\delta^3)$ where $c > 0$, this would mean that $\hat{\boldsymbol{\theta}}^T \mathbf{H}_{\mathcal{F}^*}(\boldsymbol{\theta}^*)\hat{\boldsymbol{\theta}} = -2c < 0$ and therefore that it is a strict saddle point. By considering the direction of perturbation $\hat{\boldsymbol{\theta}} = (\mathbf{I}, \mathbf{0}, \dots, \mathbf{0})$, we have

$$
\mathcal{F}^*(\boldsymbol{\theta}^* + \delta\hat{\boldsymbol{\theta}}) = \mathcal{F}^*(\delta\mathbf{I}, \mathbf{W}_{L-1}, \dots, \mathbf{W}_1)
\tag{65}
$$

$$
= \sum_{i=1}^{N} \mathbf{y}_i^T \left( \mathbf{I} + \delta^2 \left( \mathbf{I} + \sum_{\ell=2}^{L-1} \mathbf{W}_{L-1:\ell} \mathbf{W}_{L-1:\ell}^T \right) \right)^{-1} \mathbf{y}_i.
\tag{66}
$$

Denoting by $\mathbf{A} := \mathbf{I} + \sum_{\ell=2}^{L-1} \mathbf{W}_{L-1:\ell} \mathbf{W}_{L-1:\ell}^T$, we have when $\delta \to 0$

$$\mathbf{S}^{-1} = (\mathbf{I} + \delta^2 \mathbf{A})^{-1} = \mathbf{I} - \delta^2 \mathbf{A} + \mathcal{O}(\delta^3). \tag{67}$$

Hence

$$\mathcal{F}^*(\delta \mathbf{I}, \mathbf{W}_{L-1}, \ldots, \mathbf{W}_1) = \sum_{i=1}^{N} \mathbf{y}_i^T (\mathbf{I} - \delta^2 \mathbf{A} + \mathcal{O}(\delta^3)) \mathbf{y}_i \tag{68}$$

$$= \sum_{i=1}^{N} \mathbf{y}_i^T \mathbf{y}_i - \delta^2 \sum_{i=1}^{L} \mathbf{y}_i^T \mathbf{A} \mathbf{y}_i + \mathcal{O}(\delta^3) \tag{69}$$

$$= \mathcal{F}^*(\mathbf{W}_L, \mathbf{W}_{L-1}, \ldots, \mathbf{W}_1) - c\delta^2 + \mathcal{O}(\delta^3), \tag{70}$$

where $c = \sum_{i=1}^{L} y_i^T \mathbf{A} y_i > 0$ because $\mathbf{A}$ is symmetric definite positive and there exists $j$ such that $y_j \neq 0$. Hence

$$\mathcal{F}^*(\boldsymbol{\theta}^* + \delta\hat{\boldsymbol{\theta}}) = \mathcal{F}^*(\boldsymbol{\theta}^*) - c\delta^2 + \mathcal{O}(\delta^3) \tag{71}$$

which concludes the proof.

### A.3.7 Flatter global minima of the equilibrated energy (linear chains)

Here we present a preliminary investigation into the minima of the equilibrated energy compared to the MSE loss. For linear chains (§A.3.5), we show that global minima of the equilibrated energy are flatter than those of the MSE loss. More precisely, the energy global minima turn out be scaled down versions of those of the loss by the same rescaling factor of the equilibrated energy (§A.3.2). This generalises the result of [18] for linear chains with a single hidden unit.

The proof has only two steps and does not require explicit calculation of the Hessian. First, we know that we are at a global minimum of loss when we perfectly fit the data $w_{L:1}x = y$, since $\mathcal{L}(w_{L:1}x = y) = 0$. This is also true of the equilibrated energy, $\mathcal{F}^*(w_{L:1}x = y) = 0$. We can check that these are critical points by seeing that the weight gradient of the loss is null, $\nabla_{\boldsymbol{\theta}} \mathcal{L}(w_{L:1}x = y) = \mathbf{0}$, which follows from the fact that the residual is zero when we perfectly fit the data. Again, the same is true of the energy, $\nabla_{\boldsymbol{\theta}} \mathcal{F}^*(w_{L:1}x = y) = \mathbf{0}$.

The second and last step is to realise that, at these minima, the terms of the energy Hessian (Eq. 59) collapse to those of a rescaled loss Hessian (Eq. 58):

$$\frac{\partial^2 \mathcal{F}^*}{\partial w_i \partial w_j}(w_{L:1}x = y) = \begin{cases} \frac{1}{s} \frac{\partial^2 \mathcal{L}}{\partial w_i \partial w_j}, & i = j = 1 \\[2mm] \frac{1}{s} \frac{\partial^2 \mathcal{L}}{\partial w_i \partial w_j}, & i = 1, \quad j > 1 \\[2mm] \frac{1}{s} \frac{\partial^2 \mathcal{L}}{\partial w_i \partial w_j}, & i, j > 1 \end{cases} \tag{72}$$

where the rescaling is the same as that of the equilibrated energy (Eq. 56). Factoring out the rescaling

$$\mathbf{H}_{\mathcal{F}^*}(w_{L:1}x = y) = \mathbf{H}_{\mathcal{L}}(w_{L:1}x = y)/s \tag{73}$$

$$\implies \mathbf{H}_{\mathcal{F}^*}(w_{L:1}x = y) < \mathbf{H}_{\mathcal{L}}(w_{L:1}x = y), \tag{74}$$

we observe that the minima of the equilibrated energy are simply a rescaled version of those of the loss. As we saw in §A.3.2, the rescaling is positive, so it follows that the global minima of the equilibrated energy are flatter or, to put it another way, PC inference has the effect of flattening the global minima of the MSE loss (at least for linear chains).

### A.4 Experimental details

Code to reproduce all the experiments is available at https://github.com/francesco-innocenti/pc-saddles. Unless otherwise stated, for all PC networks standard Euler integration with step size $dt = 0.1$ was used to run the inference dynamics to equilibrium (§2.2, Eq. 3), with the number of iterations depending on the problem.

**Theoretical energy (Figure 1).** We trained DLNs with different number of hidden layers $H \in \{2, 5, 10\}$ on standard image classification datasets (MNIST, Fashion-MNIST and CIFAR10). At every training step, we compared the total energy (Eq. 2) at the numerical inference equilibrium $\mathcal{F}|_{\Delta \mathbf{z} \approx 0}$ with the theoretical prediction (Eq. 5). The following hyperparameters were used for all networks: 300 hidden units and SGD with learning rate $\eta = 1e^{-3}$ and batch size $b = 64$. We used a second-order explicit Runge–Kutta ODE solver (Heun) with a maximum upper integration limit $T = 300$ and an adaptive Proportional-Integral-Derivative controller (absolute and relative tolerances: $1e^{-3}$) to ensure convergence of the PC inference dynamics (Eq. 3). Results were consistent across different random initialisations.

**Toy examples (Figure 2).** All networks were linear and trained on a toy regression problem using the MSE loss (Eq. 1) and energy (Eq. 2) with output $\mathbf{y} = -\mathbf{x}, \mathbf{x} \sim \mathcal{N}(1, 0.1)$. Weights were initialised close to the origin $\mathbf{W}_{ij} \sim \mathcal{N}(0, \sigma^2)$ with $\sigma \ll 1$. For the chains, the initialisation scale was chosen to be $\sigma = 5e^{-2}$, while for the wide network it was increased to $\sigma = 1e^{-1}$ in order to make escape from the saddle faster but still visible. For PC, $T = 20$ inference iterations were used for chains and 50 for the wide network. All networks were trained with SGD and batch size $b = 64$. Learning rate $\eta = 0.4$ was used for the chains and $1e^{-3}$ for the wide network. Training was stopped when it was determined that convergence had been effectively reached, to allow for intuitive visualisation of the loss dynamics.

The landscapes were sampled on the training loss or energy with a $30 \times 30$ resolution and domain $\in [-2, 2]$ for the 2-hidden node chain and $\in [-1, 1]$ for the other networks. For the wide network, the landscape was projected onto the maximum and minimum eigenvectors of the Hessian at the origin $\boldsymbol{\theta}^* = \mathbf{0}$, $f(\boldsymbol{\theta}^* + \alpha \hat{\mathbf{v}}_{\min} + \beta \hat{\mathbf{v}}_{\max})$ since as shown by [7] random directions [28] can fail to identify saddle points. The energy landscape was plotted at the numerical equilibrium $\mathcal{F}^*(\boldsymbol{\theta})$. Figure 2 displays results for an example run, and Figure 8 shows the statistics of the training and test losses as well as gradient norms for 5 random initialisations.

**Hessian eigenspectra (Figure 3-4).** For different linear network architectures, we computed the Hessian of the loss and equilibrated energy at the origin on a random batch (size $b = 64$) of a given dataset. The datasets used were (i) a toy Gaussian with 3D input and output with the same statistics used for experiments in Figure 2, (ii) MNIST and (iii) MNIST-1D [16], a procedurally generated, one-dimensional dataset smaller than MNIST with higher model discriminability. The depth, width and data dimensions of the networks tested on the Gaussian data are clear from the vignettes in Figure 3. Figure 9 shows the same results for linear chains. For MNIST and MNIST-1D, networks with $H$ hidden layers $\{1, 2, 3\}$ had $n_\ell$ widths $\{10, 10, 5\}$ and $\{100, 50, 10\}$, respectively. Note that the MNIST networks were relatively narrow to allow for tractable computation of the Hessian. The Hessian matrices for the Gaussian data were normalised between 1 and -1, and the Hessian of the energy was computed after $T = 50$ inference iterations. For the theoretical eigenspectra of the energy Hessian, we computed the eigenvalues of Eq. 8. Figures 3 and 4 show results for an example run, and we found practically indistinguishable results for different seeds. Figures 9 & 10 show a similar analysis for a zero-rank saddle covered by Theorem 3 other than the origin.

**Experiments (Figure 5-6).** For the first set of experiment, we trained and tested linear, Tanh and ReLU networks on standard image classification tasks. Networks tested on MNIST and Fashion-MNIST had 5 fully connected (FC) layers with 500 hidden units, while those trained on CIFAR-10 had a convolutional architecture consisting of 3 blocks (with a convolution and max pooling operation) followed by two FC layers (with the last one always being linear). For PC, $T = 50$ inference iterations were used. Similar to the experiments for Figure 2, all networks were initialised close to the origin $\mathbf{W}_{ij} \sim \mathcal{N}(0, \sigma^2)$ with $\sigma = 5e^{-3}$. SGD with batch size 64 and learning rate $\eta = 1e^{-3}$ was used for all networks. PC networks were trained until the training loss reached a tolerance threshold $\mathcal{L}_{\text{train}} < 1e^{-3}$. For computational reasons, the BP-trained networks were not trained until convergence. Instead, training was stopped at as many iterations as it took PC to converge. We do report the full saddle escape dynamic for the toy examples in Figure 2 and the matrix completion experiment in Figure 6. All hyperparameters except for the initialisation remained unchanged for the other (zero-rank) saddle experiment shown in Figure 12.

For the matrix completion task (Figure 6), we attempted to replicate the experiment by [19, Figure 1] as closely as possible. Networks of depth $H = 3$ and width $n_\ell = 100$ were trained with GD and learning rate $\eta = 1e^{-2}$ to fit a 10x10 matrix of rank 3. The target matrix was generated by

multiplying two i.i.d. matrices of size 10x3 with standard Gaussian entries, and 20% of these entries were masked during training. The networks trained with PC were initialised at each saddle visited by BP, which was determined numerically by computing the rank of the network map. The origin initialisation had the same scale $\sigma = 5e^{-3}$ used in the previous experiments.

## A.5 Supplementary results

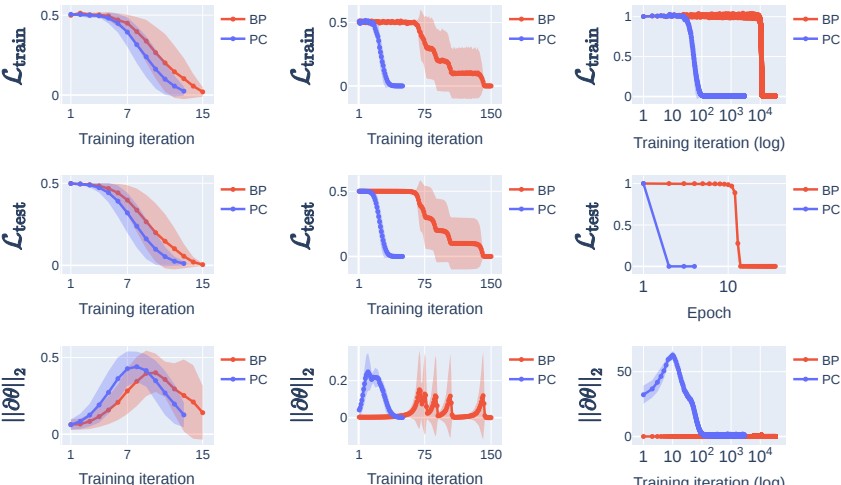

Figure 7: **Training and test statistics for linear networks of Figure 2.** For each network, we plot the mean and $\pm 1$ standard deviation of the training loss, test loss and gradient norm over 5 random initialisations. For the wide network, the test loss is evaluated once every epoch (rather than for each batch), and the training metrics are plotted on a log axis for easier visualisation. For the chain with two hidden units, the multiple loss plateaus and corresponding gradient spikes are due to different escape times from the saddle for different runs.

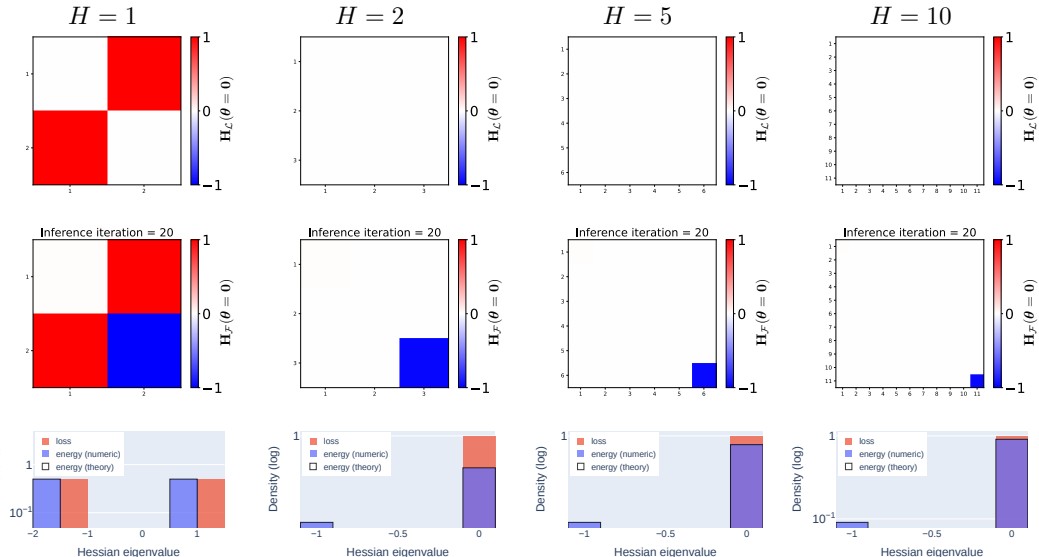

Figure 8: **Empirical verification of the Hessian at the origin of the equilibrated energy for linear chains.** This shows the same results of Figure 3 for networks of unit width $n_0 = \cdots = n_L = 1$ (see §A.4 for details). Again, we observe a perfect match between theory and experiment (see in particular Eq. 61).

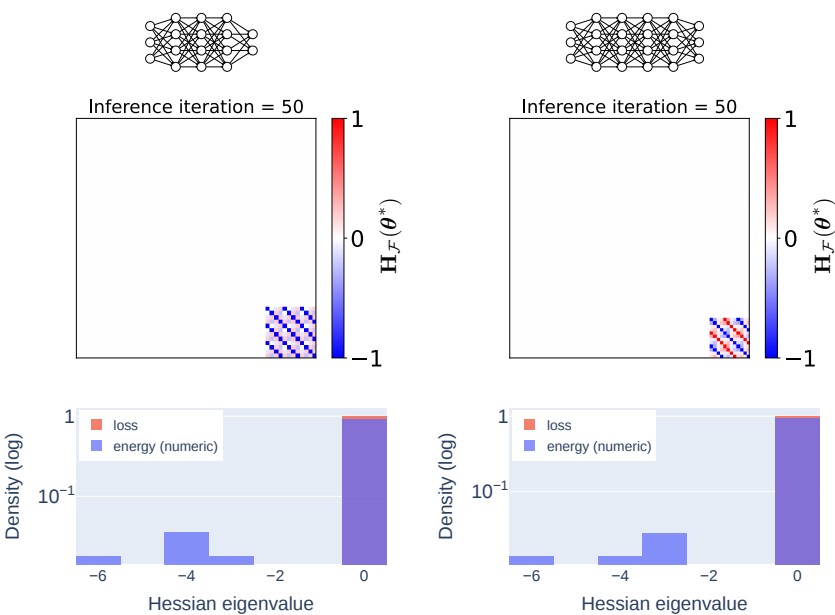

Figure 9: **Empirical verification of a strict zero-rank saddle of the equilibrated energy other than the origin for DLNs tested on a toy dataset.** We show the Hessian eigenspectrum of the MSE loss and equilibrated energy at a strict saddle other than the origin covered by Theorem 3, specifically for the critical point where all weight matrices except the penultimate are zero $\boldsymbol{\theta}^*(\mathbf{W}_\ell = \mathbf{0}, \forall \ell \neq L-1)$. We do not show the loss Hessians because they are zero for $H > 1$ (Eq. 6). The target is the same as used for Figure 3, and in the right panel one of the output dimensions is varied to be $y_2 = x_2$. Figure 10 shows results for the same critical point on MNIST and MNIST-1D.

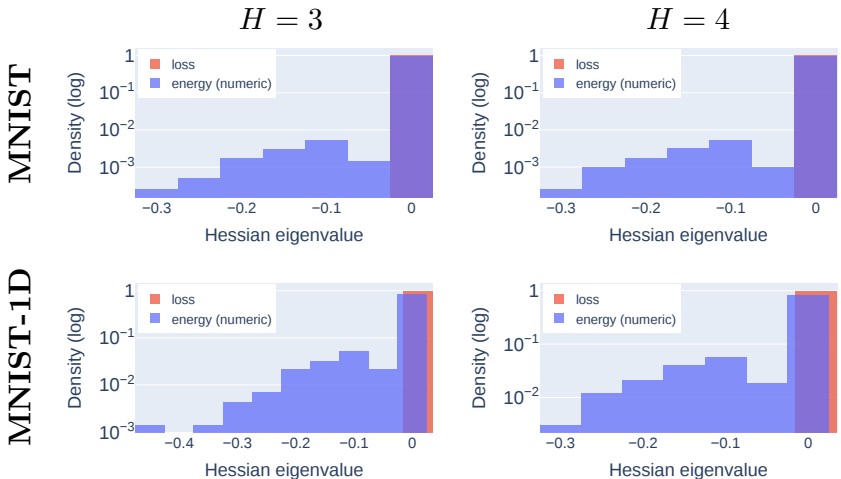

Figure 10: **Empirical verification of a strict zero-rank saddle of the equilibrated energy other than the origin for DLNs tested on more realistic datasets.** This shows similar results to Figure 9 for the more realistic datasets MNIST and MNIST-1D.

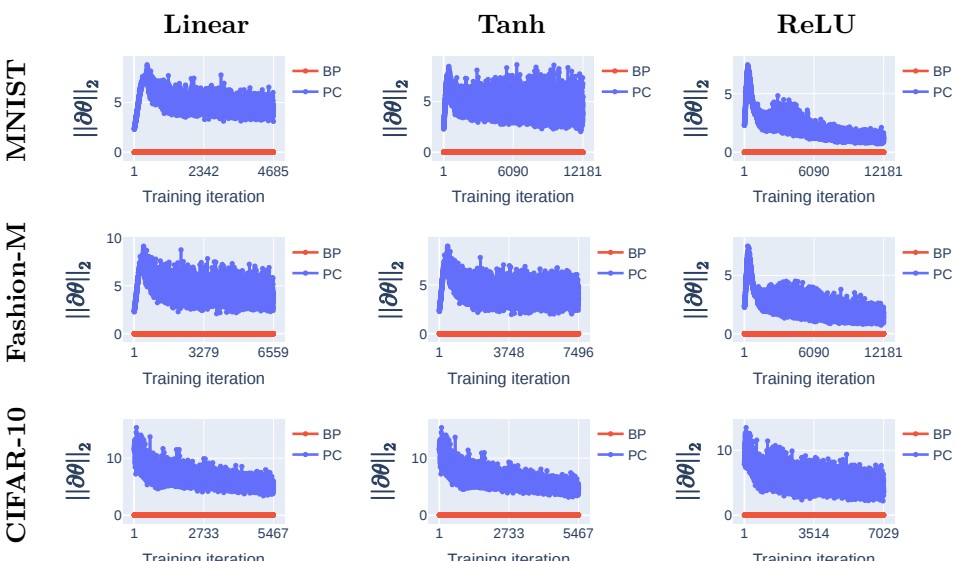

Figure 11: **No vanishing gradients for PC starting near the origin.** Weight gradient norms $||\partial\boldsymbol{\theta}||_2$ of the loss (BP) and equilibrated energy (PC) for the experiments in Figure 5.

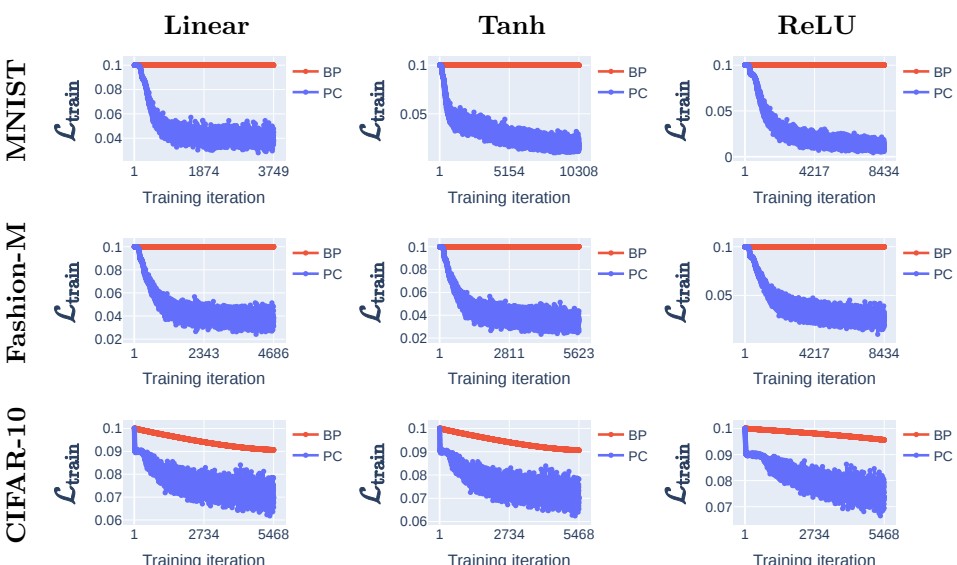

Figure 12: **PC escapes another non-strict saddle of the loss much faster than BP with SGD on non-linear networks.** This shows the same results as Figure 5 for the same saddle analysed in Figures 9 & 10 (see §A.4 for details). We show results for an example run as they were practically indistinguishable across different random seeds.

