# OpenReview forum: "Only Strict Saddles in the Energy Landscape of Predictive Coding Networks?"
_NeurIPS.cc/2024/Conference — NeurIPS 2024 poster_

### Official Review · Reviewer_KSTi · 2024-06-27

**Soundness:** 3
**Presentation:** 3
**Contribution:** 3
**Rating:** 7
**Confidence:** 3

**Summary:**

This paper studies the properties of the loss function of predictive coding (PC) networks. In PC networks, the loss function that is optimized is not a typical loss such as the MSE, but the “equilibrated energy” (or “PC energy”). The paper argues that the PC energy has some better theoretical properties than the MSE loss. Specifically, the paper conjectures that all saddle points of the PC energy are strict, in contrast with the MSE loss that has non strict saddle points. To support this conjecture, the paper compares the properties of the MSE loss function and the PC energy function around the point theta=0 in deep linear networks. In particular, they compare the spectral properties of the Hessian of these functions at theta=0. They also prove that non-strict saddle points of the MSE loss (other than theta=0) become strict in the PC energy. The paper also provides some empirical evidence that PC escapes the saddle point theta=0 of the PC energy faster than BP escapes the saddle point theta=0 of the MSE loss, even in nonlinear networks.

**Strengths:**

The paper is clearly presented.

The topic is important (PCNs and related energy-based networks deserve attention as we are looking for alternative, more energy-efficient, computing paradigms.)

The question raised is quite interesting.

Even though there is a lot of extrapolation, the argumentation to support the conjecture is well presented and sound, too.

Although the paper focuses specifically on PC networks that employ full clamping, my impression is that the analysis can likely be extended to other settings as well, with different energy functions (i.e. different “energy-based networks”), and alternative, more effective ways to do training (e.g. by nudging the output, like in Equilibrium Propagation, instead of fully clamping the output). See my questions below.

**Weaknesses:**

The theory is restricted to linear networks.

Simulations are performed only around the saddle point theta=0.

**Questions:**

1. Line 25: “PC uses only local information available to each neuron to update weights…”
Is it really the case though? To my understanding, it feels like PC needs both the transpose of the weights (in the feedback path) and knowledge of the derivative of the activation function (in nonlinear networks). It might be a very basic question, but I would like to understand what problems of BP are solved by PC networks from a hardware (neuromorphic) perspective. Could you clarify this point?

2. Is there a consensus in the deep learning literature that the strict/non-strict saddle point issue is what determines whether an optimization problem is easy/hard to solve? (It is fine if there is no consensus, it is a reasonable assumption and this question should be studied anyway).

3. Is it readily clear that the point theta=0 is a critical point of both the MSE loss and the PC energy? Is there a straightforward argument to see that the gradients of these functions at theta=0 vanish?

4. It is shown that theta=0 is a non-strict saddle point of the MSE loss. Is this also true for other loss functions? (e.g. the cross-entropy)

5. In the literature on Predictive Coding in general, it seems to be standard practice to clamp the outputs to the desired outputs. Is there a fundamental reason for doing that? Ref [1] below shows that pushing the outputs away from desired outputs (“negative nudging”) actually works significantly better than pulling them towards desired outputs (“positive nudging”) or clamping them. Note: the simulations of Ref [1] are performed on deep convolutional Hopfield networks, not PC networks, but the theory applies to PC networks too.

6. Related to Question 5 above, can the theoretical analysis performed in this work be directly transferred to the case when using negative nudging as in Ref [1] ? Or does it only work when clamping outputs to desired output values?

7. Could the analysis of the present paper be extended to other energy-based networks (not necessarily PC networks) ? Ref [2] below studies the effect of learning via Equilibrium Propagation on the Hessian of (linear) energy-based physical networks, which looks (at least superficially) similar to the theoretical study of this paper.



References

[1] Scellier, Benjamin, et al. "Energy-based learning algorithms for analog computing: a comparative study." Advances in Neural Information Processing Systems 36 (2024).

[2] Stern, Menachem, Andrea J. Liu, and Vijay Balasubramanian. "Physical effects of learning." Physical Review E 109.2 (2024): 024311.

**Limitations:**

The paper focuses on a specific feature of loss functions, namely the presence/absence of strict/non-strict saddle points. I would expect, however, that there are more important features than this one to effectively train PC networks. See my questions above.

---

> ### Author Rebuttal · Authors · 2024-08-07
>
> We thank the reviewer for their feedback and are glad that they share our excitement about the work. Below we address each point and question raised by the reviewer.
>
> The reviewer points out that our analysis of deep linear networks (DLNs) is a weakness of our theory. We would like to highlight that DLNs are the standard model for theoretical studies of the geometry of the loss landscape (see our review of related work). This is in part because of the mathematical challenges of analysing non-linear networks. Moreover, and perhaps more importantly, DLNs can be seen as a good minimal model for understanding the optimisation properties of their non-linear counterparts. While non-linearities clearly affect the geometry of the landscape, as we note in the introduction DLNs retain all the most important properties of non-linear models including non-convexity and non-linear learning dynamics.
>
> We would also like to highlight that our work goes significantly beyond previous theoretical studies on PC (reviewed in the related work section). In particular, previous theories make either non-standard assumptions or loose approximations. For example, both Alonso et al. (2023) and Innocenti et al. (2023) make second-order approximations of the energy to argue that PC makes use of Hessian information. However, our results clearly show that PC can leverage much higher-order information, turning highly degenerate, $H$-order saddles into strict ones. We now add this point of discussion to the appendix.
>
> We appreciate the point that we performed experiments only around the origin saddle. As we explain in the global rebuttal, we now include two additional sets of experiments (see attached PDF), and we refer the reviewer to that for a full explanation. In brief, the first set of experiments (Figure 13) tests another zero-rank saddle than the origin covered by Theorem 3, and the second investigates higher-rank saddles which our theory does not address (Figure 14).
>
> Whether “PC uses only local information available to each neuron to update weights” depends on what one means by “local” and “available”. We acknowledge that we were being loose with words in this particular sentence and now remove it from the paper as not critical to the main message. Nevertheless, to answer the more specific question, the reviewer is correct that standard PC needs both the transpose of the weights and the derivative of the nonlinearity, though we note that Millidge et al. (2020; https://arxiv.org/pdf/2010.01047) showed that on similar datasets one can remove these constraints without significantly harming performance. How PC could be implemented on alternative (neuromorphic) hardware is largely unknown, and we know of no published work on the topic. This is clearly a very important question for the future.
>
> The reviewer asks whether there is a consensus on how the difficulty of an optimisation problem maps onto the strict vs non-strict saddle distinction. The short answer is “yes”. We summarise the results in the introduction (paragraph 4) and provide an in-depth review in the related work section. At a very high level, strict saddles are benign for even first-order algorithms like GD as long as one does not initialise too close to them or uses a particularly small learning rate, while non-strict ones can effectively trap (S)GD in the phenomenon of vanishing gradients since they are very flat. As we note in the paper, adaptive gradient methods like Adam have been argued to better deal with vanishing gradients and curvature in this work, but and their behaviour near non-strict saddles is much less understood than that of (S)GD.
>
> The reviewer asks whether there is a straightforward way of seeing that the gradient of both the MSE and the PC energy vanish at the origin ($\theta=0$). The simplest case where this can be seen is for a network with two weights, where the loss is $\mathcal{L} = ½(y- w_2 w_1 x)^2$ and the (equilibrated) energy is $\mathcal{F}^* = \mathcal{L} / 1+w_2^2$. Because the residual $r = (y- w_2 w_1 x)$ is quadratic in the weights, the gradient is zero. Specifically, $\partial \mathcal{L} / \partial w_1 = -w_2xr$ and $\partial \mathcal{L} / \partial w_2 = -w_1xr$. For the energy, the expression of the gradient is a little more involved and has basically 2 terms. One is the gradient of the loss scaled by $1+w_2^2$ which will therefore also vanish at 0. The other term depends on the derivative of the rescaling, in this case $\partial (1+w_2^2)/ \partial w_i$. The rescaling does not depend on the first weight so this vanishes, and for the second weight is also zero because it is again quadratic in that weight.
>
> The reviewer asks whether the origin can be a non-strict saddle for other loss functions than the MSE. This is an interesting point, and it turns out to be true for any generic convex, twice-differentiable cost as soon as we have two or more hidden layers as shown by Jacot et al. (2021, last paragraph before Section 5.1; https://arxiv.org/pdf/2106.15933).
>
> The reviewer asks about the relationship between PC and equilibrium propagation (EQ). As shown by Millidge et al. (2022; https://arxiv.org/pdf/2206.02629), PC can be seen as an EQ algorithm where the free phase is minimised with a feedforward sweep (i.e.  the squared energy of the last layer is equivalent to the nudging term). This avoids the need for two phases and is arguably more biologically plausible. It is not clear to us whether the theory could be generalised to the case of negative nudging or other energy-based networks, but this could be an interesting future direction. We thank the reviewer for directing us to reference [2] which we agree could potentially open interesting connections.

---

> > ### Comment · Reviewer_KSTi · 2024-08-12
> >
> > I thank the authors for their detailed reply.
> >
> > You explain in your rebuttal that there is currently no proposal of neuromorphic implementations of PC networks in the literature. I have one more question: how do you envision PC networks being useful? Do you see PC networks being useful as a theory of computational neuroscience? Or do you see them being useful for machine learning?
> >
> > Unless you think the analysis of your work is strictly restricted to PCNs, I’d encourage you to write a paragraph in the discussion section about the broader field of energy-based networks (including analog/physical energy-based networks, see e.g. Refs [1,2] above), and whether your theory could be transposed to such networks.

---

> > > ### Author Response · Authors · 2024-08-12
> > >
> > > We believe that PC will likely continue to be a productive theoretical framework for neuroscience and cognitive science, and it could perhaps also help machine learning to address current limitations such as inference-learning trade-offs and energy efficiency. We think that the bio-inspired learning community working on alternatives to backprop broadly shares this belief.
> > >
> > > We will include a paragraph discussing the potential implications of this work for the broader field of physical energy-based networks including the references pointed out by the reviewer. To expand on our previous answer, it is not clear to us whether the theory can be easily transferred to other energy-based algorithms like EQ because it depends on whether they optimise the same equilibrated energy we derive and characterise the properties of. Nevertheless, Ref. [2] clearly shows that the inference dynamics of some energy-based systems change the Hessian of the weights in beneficial ways. This makes us think that this principle (of reshaping the weight landscape for easier learning) could be quite general, and the question might be how different energy-based algorithms change the landscape. We are excited about this direction, and it is indeed one of the motivations behind our work given the recent connections between PC and other energy-based systems.

---

### Official Review · Reviewer_cEwT · 2024-07-09

**Soundness:** 3
**Presentation:** 4
**Contribution:** 3
**Rating:** 7
**Confidence:** 3

**Summary:**

This paper explores how the energy landscape in deep linear networks trained with predictive coding compares to the loss landscape of DLNs trained with backpropogation. The authors prove that the energy of PC is a rescaled version of the MSE loss. The authors then show that, unlike BP trained DLNs, multiple points in the energy landscape of strict saddles. This suggests a way in which training with PC can escape poor initializations faster than BP. To support these theoretical results, the authors provide numerical results.

**Strengths:**

1. The paper was well written and well motivated. It was clear what the problem the authors were tackling was, why it was important, and what the main results of the paper were.

2. The combination of theoretical and numerical results that collectively contributed to a strong, compelling story was great.

**Weaknesses:**

Major C1: Given that the authors prove that, for DLNs, the equilibriated energy is a scaled version of the MSE, I believe an alternative explanation for why training with PC is able to achieve good performance faster than BP is because it has a different effective learning rate. Is that assessment correct? If so, did the authors explore whether changing the learning rate for BP led to changes in learning speed in the DLN and DNN numerical examples?

Major C2: While there is technical discussion of PC in Section 2, it was not clear how that maps to the idea of PC in neuroscience. More discussion, even in the Appendix, would be useful for those familiar with PC more from the neuroscience perspective.

Minor C1: It was unclear to me why $\textbf{W}_L$ had to equal 0 for the equilibrated energy (line 235). I read the subsection in the Appendix that is referenced and still felt like the comment was a little abrupt.

Minor C2: Given that the authors have some remaining space, it would be interesting and good to hear more about the neuroscience perspective, other than learning speed.

Very minor C1: $\textbf{g}_f$ is used before it is defined (line 154).

Very minor C2: Why is there a 1/2 in the MSE loss? To the best of my knowledge, this is not "standard" as is quoted in line 160.

**Questions:**

1. Does PC still show an advantage over BP if a different learning rate is used for BP (so as to match the effective learning rate of PC)?

2. Why must $\textbf{W}_L =0$ for the saddles considered at the end of the paper?

3. Why would having strict saddles be useful in cognitive science?

**Limitations:**

The authors did a nice job addressing their limitations.

---

> ### Author Rebuttal · Authors · 2024-08-07
>
> We thank the reviewer for their feedback. Below we address each point raised by the reviewer.
>
> Given that the equilibrated energy turns out to be a rescaled version of the MSE loss, the reviewer asks if PC could be interpreted as having a different effective learning rate than BP. Though pointing in the right direction, we think that this is an insufficient explanation. First, the rescaling depends on the weights and so it will change at every weight update. In this sense, the gradient magnitude (or effective learning rate) of PC could be seen as adaptive, specifically higher for small weights and smaller for bigger weights. This is easy to see for the simple case of a network with two weights, where the equilibrated energy is just $\mathcal{L}/1+w_2^2$. More to the point, however, the rescaling not only affects the magnitude of the gradient, but also the curvature and other higher-order derivatives of the energy. To see this, consider an initialisation near the origin, where the weights are approximately 0 as can be seen from the toy model above, and the rescaling is therefore close to the identity or 1, $\mathcal{F} \approx \mathcal{L}$. If PC had the same effective learning rate as BP, then it would also show slow training dynamics near the origin (for networks with more than one hidden layer where the saddle is non-strict). However, this is not what we observe, theoretically and empirically.
>
> Changing the learning rate of BP to match the “effective learning rate” of PC would therefore not work, in the sense that there is no learning rate for which the two algorithms will show exactly the same (S)GD dynamics. Nevertheless, we agree with the reviewer that it is an interesting question whether higher learning rates would allow BP with SGD to escape the saddle faster than PC. However, as we explain in our global response, such an analysis was beyond the scope of this paper, and we refer the reviewer to that rebuttal for our argument.
>
> We agree with the reviewer that we did not particularly motivate our work from a neuroscience perspective and we discussed only superficially its implications for the field. There are many excellent reviews of PC we cite (the most recent being Salvatori et al., 2023) in the first sentence of the paper that cover the neuroscientific origins and connections of PC.
>
> We now expand on the neuroscience implications of our work based on Song et al. (2024; https://www.nature.com/articles/s41593-023-01514-1), who proposed that energy-based algorithms like PC rely on a fundamentally different principle of credit assignment for learning in the brain called “prospective configuration”—where essentially weights follow activities (rather than the other way around). Based on a range of empirical results, Song et al. (2024) argued that PC confers many benefits for learning including faster convergence. Our theoretical results suggest that this latter claim may have been overstated. In particular, our results clearly show that PC and BP operate on two different landscapes and that therefore convergence speed will depend on many factors including network depth, width, dataset (specifically output covariance), initialisation, learning rate, and optimiser–all of which affect the landscape geometry or the learning dynamics. Any universal claim about faster convergence of PC should therefore control for all these factors. Song et al. (2024) claimed that PC learns faster than BP mainly based on an experiment with a deep ($L=15$) but narrow ($n_\ell=64$) network controlling for learning rate. However, our work reveals the importance of the initialisation of the weights, which interacts non-trivially with the network width. In particular, Orvieto et al. (2022) showed, in brief, that the narrower the network, the closer one will start from the origin saddle for standard initialisations. This insight likely explains the speed-up observed by Song et al. (2024). We add the above point to the discussion and thank the reviewer for highlighting this deficiency and allowing us to improve the reach of the paper.
>
> We agree with the reviewer that it is not very clear why $W_L = 0$ in order for the gradient of the equilibrated energy to be zero. We can see no simple or intuitive explanation, as it simply turns out to be a property of the energy gradient. In particular, this is because of the rescaling term $S = I_ +W_LW_L^T + …$, which as we highlight contains a term quadratic in the last weight matrix. The derivative of the rescaling $\partial S / \partial W_L$ needs to be zero in order for the gradient of the equilibrated energy to be zero. Because as just noted the rescaling is quadratic in $W_L$, the derivative will be linear in $W_L$ and so the last weight matrix will have to be zero for the gradient to be zero. We now clarify and expand on the explanation of this property in the appendix, which as pointed out by the reviewer was previously only noted as a side remark.
>
> The gradient $g_f$ is now defined before being used, and we thank the reviewer for pointing out this imprecision.
>
> The ½ in the MSE loss (and energy) is often used as a theoretical convenience to cancel out the factor of 2 when taking derivatives of squared losses. It does not affect any of the derivations or theoretical results.

---

> > ### Comment · Reviewer_cEwT · 2024-08-07
> >
> > I thank the authors for their detailed rebuttal. I better understand now the goals of the paper (as the authors have re-stated them in the global rebuttal), as well as better understand why a larger learning rate is not a sufficient explanation for the success of PC. I additionally like the new experiments on matrix completion, which provides a nice new demonstration of the difference in loss landscape between PC and BP.
> >
> > Given that my score was already high, I have decided to keep it as is. However, I now feel more confident in the quality of the work and believe more strongly that it should be accepted. I will make this clear to the Area Chair in the post-discussion period.

---

### Official Review · Reviewer_hPaX · 2024-07-13

**Soundness:** 3
**Presentation:** 3
**Contribution:** 2
**Rating:** 5
**Confidence:** 4

**Summary:**

This work looks at the (equilibrated) loss landscape of predictive coding networks, and analyzes the nature of saddle points therein --- especially in comparison to that obtained for backpropagation based deep linear networks with MSE loss. They find that several non-strict saddle points in the latter turn to strict saddle points (with positive and negative eigenvalues) in the former. This is argued to lead to better convergence properties, and is shown empirically for some simple networks.

**Strengths:**

- Studying the loss landscapes for predictive coding is a very interesting direction, and could be useful to better understand the benefits imparted by it, and also circumvent any idiosyncratic problems that might arise.

- An interesting theoretical analysis of the Hessian for predictive coding is carried out, by considering the equilibrated loss, which can also be useful for future theoretical works in this space.

- Non-strict saddles at origin or zero-rank saddles are shown as strict saddles in predictive coding, which afford for better convergence when training starts near such a saddle as compared to that with usual deep nets where these are non-strict saddles. The experiments support the point and associated visualizations are quite interesting.

**Weaknesses:**

Some of the claims might a bit exaggerated: In the conclusion, the authors write that PC networks have only strict saddles. But, I don't think there is anywhere in the paper it is shown that there are only two kinds of saddles: zero and zero-rank, both of which are shown as strict saddles. No other saddles are analyzed, or shown to be non-existent, and the zero-rank saddle is not too dissimilar to the zero saddle. Likewise, there are statements saying that they prove non-strict saddles other than origin to be zero in PC network setting, but I think they are just talking about the zero-rank saddles.

Experiments are somewhat restricted: The experiments are primarily with simple MLPs. It is also unclear when the backprop'ed networks are trained if their convergence results are well-tuned. In other words, if they use GD vs SGD, or if the learning rates or batch sizes are optimal, adaptive gradient methods, and the kind. (PC doesn't have to outperform them, but it will be good to gauge how well it fares in context of all these things that practitioners would normally employ)

Section 3.2 analysis is more or less a corollary of Singh et al, 2021: The authors only somewhat acknowledge this in the appendix, but I think this should be properly highlighted and attributed in the main text of the paper as well.

Loss Landscape of PCNs could have been studied a bit better: The technical part is largely based on Achour et al and Singh et al, and some more aspects of the loss landscape could have been carried out. The paper has a good contribution, but I am unsure if this is sufficient compared to a normal NeurIPS paper of 9 pages.

Others:
- Some of the terminology is loose and confusing: Often in their paper the talk about saddles of BP in reference to that of MSE. I think this is quite confusing, as what they mean by MSE is networks trained with Backprop on MSE. A better acronym would be BP-MSE or BP, as otherwise it is quite vague.
- Line 49, Phenomenon of vanishing gradients has the wrong reference. The right one is Bengio 1994 https://www.comp.hkbu.edu.hk/~markus/teaching/comp7650/tnn-94-gradient.pdf, while the currently mentioned one is more about curvature.
- Could the authors better describe the steps from eqn 70 to 71 in proof of proposition 1, and where all things are under approximation and where equality? It's a bit loosely written at the moment.

**Questions:**

See the above section

**Limitations:**

^^

---

> ### Author Rebuttal · Authors · 2024-08-07
>
> We thank the reviewer for their feedback. Below we address the points made by the reviewer one by one.
>
> We tried to be very careful to not overstate our claims based on the results. In the abstract, introduction and conclusion, we state that “we provided theoretical and empirical evidence that the effective energy landscape of PC networks has only strict saddles”. Nevertheless, we agree with the reviewer that this could be misinterpreted to overstate our results, so we removed this sentence from the abstract and edited it more conservatively in the introduction and discussion. To avoid overstating our results, we also note that throughout the paper we used word *conjecture* to emphasise that we do not prove that all the saddles of the equilibrated energy are strict. For the same reason, we included a question mark in the title of the paper.
>
> The reviewer is correct that when we say “other non-strict saddles than the origin” we are referring to the zero-rank saddles. It is true that the origin is one of these saddles as we point out. However, there is no a priori reason to believe that these zero-rank saddles (other than the origin) should also be strict in equilibrated energy. Nevertheless, we agree with the reviewer that we did not study other types (higher-rank) of saddles, theoretically or empirically. Therefore, as we explain in our global response, we now also include two additional sets of experiments (in the attached PDF), one looking at another zero-rank saddles than the origin and the other supporting the strictness of higher-rank saddles of the equilibrated energy.
>
> The reviewer points out that our “experiments are primarily with simple MLPs”. This is true, although we note that we also performed experiments on convolutional networks too as noted in the caption of Figure 5.
>
> We completely agree with the reviewer that the convergence results were not at all tuned. As we explain in our global rebuttal, our main goal was to characterise the intrinsic geometry of the equilibrated energy landscape, which had not been done before. Studying under which conditions, such different learning rates and optimisers, the energy landscape might be faster to optimise than the loss landscape is beyond the scope of our work. We refer to the global rebuttal for our argument and note that our work facilitates the study of such conditions.
>
> We strongly disagree that Section 3.2 is a corollary of Singh et al. (2021). Singh et al. (2021) derived expressions for the Hessian blocks of arbitrary deep linear networks with MSE loss. They did not focus on characterising the landscape. We simply use the notation employed by Singh et al. (2021) to derive the Hessian of the equilibrated energy, which is non-trivial and significantly goes beyond this work. The reason why we include a re-derivation of the Hessian (with slightly different notation) in the appendix, and its structure at the origin in section 3.2 (which as we note was first derived by Kawaguchi, 2016, with a different Hessian derivation) is for intuitive comparison with that of the equilibrated energy.
>
> We agree with the reviewer that we do not provide a full picture of the landscape, including all saddles and minima. However, we strongly disagree that “the technical part is largely based on Achour et al and Singh et al”. As explained above, we use mainly the notation of Singh et al. and re-derive the Hessian of the loss for comparison only. Achour et al. characterised all the critical points of the MSE for DLNs to second order, and we simply use their characterisation for comparison (i.e. to check what happens to these critical points in the equilibrated energy). Importantly, we would like to emphasise that showing whether the critical points of the MSE loss (characterised by Achour et al.) are also critical points of the energy and, if so, what kind of points, are additional questions that require a completely separate technical analysis of the gradient and Hessian of the equilibrated energy, which as our calculations show are non-trivial to derive. As we also review in the related work section, it is worth emphasising that the landscape of DLNs for the MSE loss took many papers before being understood to the degree that it is today. Our paper can be seen as a first step into understanding the landscape of the equilibrated energy. Indeed, our novel derivation of the equilibrated energy enables further studies into the landscape.
>
> We agree with the confusions in terminology pointed out by the reviewer and made changes to the text accordingly.
>
> We agree with the reviewer that the Bengio et al. (1994) paper is the first to introduce the vanishing gradient problem. We made reference to the other paper (Orvieto et al., 2022) because this was the first (to the best of our knowledge) to connect the phenomenon of vanishing gradients to vanishing curvature and therefore the non-strict saddle at the origin addressed in part by our work. We now refer to both papers to also acknowledge Bengio et al. (1994).
>
> We agree with the reviewer that our original proof of Theorem 3 was loosely written. We now update the paper with a full proof including all steps and clearly separating the equalities from the approximations. We thank the reviewer for pointing out this imprecision.

---

> ### Comment · Reviewer_hPaX · 2024-08-14
>
> Thanks for the rebuttal. Please be more precise in the claims in the text about the kind of results proved. Also, don't take me wrong in comparisons with Achour/Singh et al. Please see the qualifications in my original comments. Then there is definitely novelty in looking at the landscape of equilibrated energy, my comment is more on eqns 6 and 7 and not 3.2 as a whole and then it's broadly the way the discussion comes off in the main text. And, likewise, for the discussion around Achour et al. Just make it clear what is known from before, what is evident from prior work even if unsaid, and what you are introducing.

---

> > ### Author Response · Authors · 2024-08-14
> >
> > We will further clarify our claims and what is known vs what we introduce throughout the text. Regarding equations 6 and 7, we now add a further clarification that we are presenting known results on the loss purely for intuitive comparison with the equilibrated energy.

---

### Author Rebuttal · Authors · 2024-08-06

We thank all of the reviewers for taking the time to read our paper and their feedback, which we believe improved the clarity, scope and contributions of the paper. In this global response, we address points that were raised by more than one reviewer and outline other relatively minor changes we made to the paper to address specific points made by individual reviewers.

We identified two important, related concerns raised by the reviewers:
1. **Whether the reported faster SGD convergence of PC when initialised near the origin (Figure 5) holds under different hyperparameters, such as learning rates and optimisers (e.g. Adam).**

It was beyond the scope of this work—which was mainly concerned with the intrinsic geometry of the (equilibrated) energy landscape—to address this issue, and we completely agree with the reviewers that the original writing of the paper gave the (wrong) impression that this was one of our aims. We now make our goal clear throughout the paper. As we now explain in the discussion, studying all the different conditions under which the energy landscape might be faster to optimise than the loss landscape is a difficult question requiring further study. This is for a few reasons. First, one can always make the initialisation scale small enough such that BP will not escape (even with Adam) one of the considered saddles to show that PC is faster. The question then becomes whether this will have benefits for performance or generalisation and whether there might be a trade-off with learning speed. Second, the behaviour of adaptive gradient optimisers like Adam near non-strict saddles is not as well understood as that of (S)GD. Finally, characterising the geometry of minima of the equilibrated energy—which our work enables—would also be probably necessary to understand the overall convergence behaviour of PC.

We also highlight that our theoretical results do not necessarily suggest that PC will always be faster. In fact, we now argue that our results explain the speed-ups observed by Song et al. (2024; https://www.nature.com/articles/s41593-023-01514-1) on a deep network because of the small network width used, which as we explain in response to one reviewer determines how closely one starts from the origin for standard initialisations (as used in that paper).

2. **Limited experiments.**

Reviewers pointed out that our experiments were quite limited. This was in part motivated by the above concern about the impact of different hyperparameters on convergence speed, which as we hope we now clarified is beyond the scope of our work.

Nevertheless, setting aside the question of hyperparameter tuning, we acknowledge that we performed experiments on non-linear networks only when initialising close to the origin saddle. For that reason, we now include two additional sets of experiments (see attached PDF). The first (attached Figure 13) is simply a replica of the experiments in Figure 5 for another non-strict zero-rank saddle of the MSE that we proved to be strict for the equilibrated energy (Theorem 3). We also note that our submitted code makes it easy to test for other zero-rank saddles by simply changing the initialisation.

The second set of experiments (attached Figure 14) numerically investigates the question of higher-rank saddles which our theory did not address and was a specific concern raised by one of the reviewers. We compared the training loss of BP and PC with GD on a matrix completion task studied by Jacot et al. (2021, Figure 1; https://arxiv.org/pdf/2106.15933), where as they show if one starts near the origin, GD visits a sequence of saddles, each representing a solution of higher rank. As the attached Figure 14 shows, PC quickly escapes all the saddles (specifically of rank 0, 1 and 2) visited by BP. We refer to the Figure caption for more details. These results go beyond our theory (restricted to zero-rank saddles) to support our general conjecture that all the saddles of the equilibrated energy are strict.

Besides generally improving the clarity of the writing, we made other minor changes to address points made by individual reviewers:
* We emphasise the contribution of our theoretical result of the equilibrated energy as a rescaled MSE loss. We argue that this is important because (i) it corrects a previous mistake in the literature that the MSE loss is equal to the output energy as we now discuss in the paper, and (ii) it enables further studies into the energy landscape (e.g. minima) as we note in the discussion.
* We clarify and expand on the implications of the paper. We now include the most important implications and limitations in the abstract and expand on them in the discussion. In particular, we emphasise the point–which was previously only secondary–that the strictness of the origin saddle makes PC more robust to vanishing gradients, providing supporting plots of the norms of the weight gradients for all the experiments. In addition, we discuss how our results suggest that previous claims of faster convergence of PC may have been overstated and highlight the limitation of scaling PC to deeper architectures where inference becomes difficult to solve.
* To make space for the additional experiments and expanded discussion, we moved the related work section to the appendix.
* In the introduction, we better motivate the use of deep linear networks as our theoretical model, which is the standard for studies of the loss landscape geometry. We also explain how our analysis goes beyond most previous theoretical works on PC and elaborate on this point in the related work section.
* We fix some imprecise or unclear notations and derivations pointed out by reviewers. We also provide an expanded proof of the strictness of the zero-rank saddles (Theorem 3) in the appendix as requested by one reviewer.

---

### Decision · Program_Chairs · 2024-09-25

**Decision:**

Accept (poster)

**Comment:**

This paper studies the optimization landscape of predictive coding for training deep linear networks. The authors prove that the energy of predictive coding is a rescaled version of the MSE loss and then show that several non-strict saddle points in standard training become strict saddle points in predictive coding training. This could potentially lead to better convergence.

Studying the loss landscapes for predictive coding could be useful in better understanding the benefits of predictive coding. However, as pointed out by reviewer, this paper only studies two kinds of saddles—zero and zero-rank—not all saddle points, which should be clearly stated.